# Combining data and theory for derivable scientific discovery with AI-Descartes

Cristina Cornelio [1,2] ✉, Sanjeeb Dash[1], Vernon Austel[1], Tyler R. Josephson [3,4], Joao Goncalves[1], Kenneth L. Clarkson[1], Nimrod Megiddo[1], Bachir El Khadir[1] & Lior Horesh [1,5] ✉

Scientists aim to discover meaningful formulae that accurately describe experimental data. Mathematical models of natural phenomena can be manually created from domain knowledge and fitted to data, or, in contrast, created automatically from large datasets with machine-learning algorithms. The problem of incorporating prior knowledge expressed as constraints on the functional form of a learned model has been studied before, while finding models that are consistent with prior knowledge expressed via general logical axioms is an open problem. We develop a method to enable principled derivations of models of natural phenomena from axiomatic knowledge and experimental data by combining logical reasoning with symbolic regression. We demonstrate these concepts for Kepler's third law of planetary motion, Einstein's relativistic time-dilation law, and Langmuir's theory of adsorption. We show we can discover governing laws from few data points when logical reasoning is used to distinguish between candidate formulae having similar error on the data.

Artificial neural networks (NN) and statistical regression are commonly used to automate the discovery of patterns and relations in data. NNs return "black-box" models, where the underlying functions are typically used for prediction only. In standard regression, the functional form is determined in advance, so model discovery amounts to parameter fitting. In symbolic regression (SR)[1,2], the functional form is not determined in advance, but is instead composed from operators in a given list (e.g., $+$, $-$, $\times$, and $\div$) and calculated from the data. SR models are typically more "interpretable" than NN models, and require less data. Thus, for discovering laws of nature in symbolic form from experimental data, SR may work better than NNs or fixed-form regression[3]; integration of NNs with SR has been a topic of recent research in neuro-symbolic AI[4–6]. A major challenge in SR is to identify, out of many models that fit the data, those that are scientifically meaningful. Schmidt and Lipson[3] identify meaningful functions as those that balance accuracy and complexity. However many such expressions exist for a given dataset, and not all are consistent with the known background theory.

Another approach would be to start from the known background theory, but there are no existing practical reasoning tools that generate theorems consistent with experimental data from a set of known axioms. Automated Theorem Provers (ATPs), the most widely-used reasoning tools, instead solve the task of proving a conjecture for a given logical theory. Computational complexity is a major challenge for ATPs; for certain types of logic, proving a conjecture is undecidable. Moreover, deriving models from a logical theory using formal reasoning tools is especially difficult when arithmetic and calculus operators are involved (e.g., see the work of Grigoryev et al.[7] for the case of inequalities). Machine-learning techniques have been used to improve the performance of ATPs, for example, by using reinforcement learning to guide the search process[8]. This research area has received much attention recently[9–11].

[1]IBM Research—Mathematics and Theoretical Computer Science, New York, NY, USA. [2]Samsung AI—Machine Learning and Data Intelligence, Cambridge, UK. [3]Department of Chemical, Biochemical, and Environmental Engineering, University of Maryland, Baltimore County, MD, USA. [4]Department of Chemistry and Chemical Theory Center, University of Minnesota, Minneapolis, MN, USA. [5]Columbia University, Computer Science, New York, NY, USA. ✉ e-mail: c.cornelio@samsung.com; lhoresh@us.ibm.com

Models that are derivable, and not merely empirically accurate, are appealing because they are arguably correct, predictive, and insightful. We attempt to obtain such models by combining a novel mathematical-optimization-based SR method with a reasoning system. This yields an end-to-end discovery system, which extracts formulas from data via SR, and then furnishes either a formal proof of derivability of the formula from a set of axioms, or a proof of inconsistency. We present novel measures that indicate how close a formula is to a derivable formula, when the model is provably non-derivable, and we calculate the values of these measures using our reasoning system. In earlier work combining machine learning with reasoning, Marra et al.[12] use a logic-based description to constrain the output of a GAN neural architecture for generating images. Scott et al.[13] and Ashok et al.[14] combine machine-learning tools and reasoning engines to search for functional forms that satisfy prespecified constraints. They augment the initial dataset with new points in order to improve the efficiency of learning methods and the accuracy of the final model. Kubalik et al.[15] also exploit prior knowledge to create additional data points. However, these works only consider constraints on the functional form to be learned, and do not incorporate general background-theory axioms (logic constraints that describe the other laws and unmeasured variables that are involved in the phenomenon).

## Results

### Discovery as a formal mathematical problem

Our automated scientific discovery method aims to discover an unknown *symbolic model* $y = f^*(\mathbf{x})$ (bold letters indicate vectors) where $\mathbf{x}$ is the vector $(x_1, \ldots, x_n)$ of independent variables, and $y$ is the dependent variable. The discovered model $f$ (an approximation of $f^*$) should fit a collection of $m$ data points, $((\mathbf{X}^1, Y^1), \cdots, (\mathbf{X}^m, Y^m))$, be derivable from a background theory, have low complexity and bounded prediction error. More specifically, the inputs to our system are 4-tuples $\langle \mathcal{B}, \mathcal{C}, \mathcal{D}, \mathcal{M} \rangle$ as follows.

- Background Knowledge $\mathcal{B}$: a set of domain-specific axioms expressed as logic formulae. They involve $\mathbf{x}$, $y$, and possibly more variables that are necessary to formulate the background theory. In this work we focus mainly on first-order-logic formulae with equality, inequality and basic arithmetic operators. We assume that the background theory $\mathcal{B}$ is *complete*, that is, it contains all the axioms necessary to comprehensively explain the phenomena under consideration, and *consistent*, that is, the axioms do not contradict one another. These two assumptions guarantee that there exists a unique derivable function $f_\mathcal{B}$ that logically represents the variable of interest $y$. Note that although the derivable function is unique, there may exist different functional forms that are equivalent on the domain of interest. Considering the domain with two points $\{0, 1\}$ for a variable $x$, the two functional forms $f(x) = x$ and $f(x) = x^2$ both define the same function.
- A Hypothesis Class $\mathcal{C}$: a set of admissible symbolic models defined by a grammar, a set of invariance constraints to avoid redundant expressions (e.g., $A + B$ is equivalent to $B + A$) and constraints on the functional form (e.g., monotonicity).
- Data $\mathcal{D}$: a set of $m$ examples, each providing certain values for $x_1, \ldots, x_n$, and $y$.
- Modeler Preferences $\mathcal{M}$: a set of numerical parameters (e.g., error bounds on accuracy).

### Generalized notion of distance

In general, there may not exist a function $f \in \mathcal{C}$ that fits the data exactly and is derivable from $\mathcal{B}$. This could happen because the symbolic model generating the data might not belong to $\mathcal{C}$, the sensors used to collect the data might give noisy measurements, or the background knowledge might be inaccurate or incomplete. To quantify the compatibility of a symbolic model with data and background theory, we introduce the notion of *distance* between a model $f$ and $\mathcal{B}$. Roughly, it reflects the error between the predictions of $f$ and the predictions of a formula $f_\mathcal{B}$ derivable from $\mathcal{B}$ (thus, the distance equals zero when $f$ is derivable from $\mathcal{B}$). Figure 1 provides a visualization of these two notions of distance for the problem of learning Kepler's third law of planetary motion from solar-system data and background theory.

### Integration of statistical and symbolic AI

Our system consists mainly of an SR module and a reasoning module. The SR module returns multiple candidate symbolic models (or formulae) expressing $y$ as a function of $x_1, \ldots, x_n$ and that fit the data. For each of these models, the system outputs the distance $\varepsilon(f)$ between $f$ and $\mathcal{D}$ and the distance $\beta(f)$ between $f$ and $\mathcal{B}$. We will also be referring to $\varepsilon(f)$ and $\beta(f)$ as errors.

These functions are also tested to see if they satisfy the specified constraints on the functional form (in $\mathcal{C}$) and the modeler-specified level of accuracy and complexity (in $\mathcal{M}$). When the models are passed to the reasoning module (along with the background theory $\mathcal{B}$), they are tested for derivability. If a model is found to be derivable from $\mathcal{B}$, it is returned as the chosen model for prediction; otherwise, if the reasoning module concludes that no candidate model is derivable, it is necessary to either collect additional data or add constraints. In this case, the reasoning module will return a quality assessment of the input set of candidate hypotheses based on the distance $\beta$, removing models that do not satisfy the modeler-specified bounds on $\beta$. The distance (or error) $\beta$ is computed between a function (or formula) $f$, derived from numerical data, and the derivable function $f_\mathcal{B}$ which is implicitly defined by the set of axioms in $\mathcal{B}$ and is logically represented by the variable of interest $y$. The distance between the function $f_\mathcal{B}$ and any other formula $f$ depends only on the background theory and the formula $f$ and not on any particular functional form of $f_\mathcal{B}$. Moreover, the reasoning module can prove that a model is not derivable by returning counterexample points that satisfy $\mathcal{B}$ but do not fit the model.

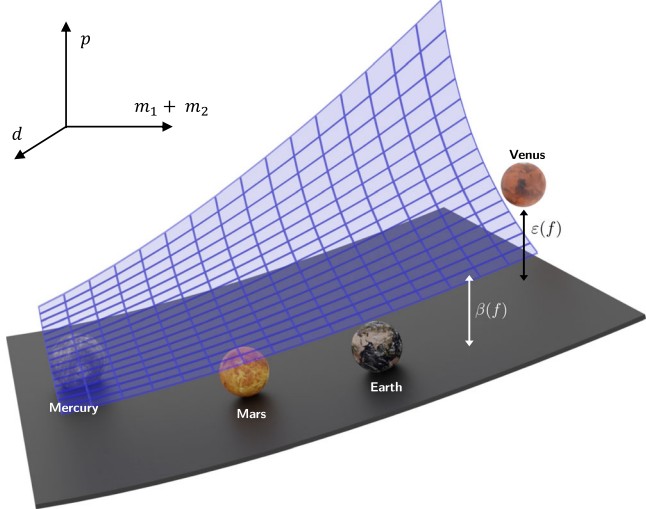

**Fig. 1 | Visualization of relevant sets and their distances.** The numerical data, background theory, and a discovered model are depicted for Kepler's third law of planetary motion giving the orbital period of a planet in the solar system. The data consists of measurements $(m_1, m_2, d, p)$ of the mass of the sun $m_1$, the orbital period $p$ and mass $m_2$ for each planet and its distance $d$ from the sun. The background theory amounts to Newton's laws of motion, i.e., the formulae for centrifugal force, gravitational force, and equilibrium conditions. The 4-tuples $(m_1, m_2, d, p)$ are projected into $(m_1 + m_2, d, p)$. The blue manifold represents solutions of $f_\mathcal{B}$, which is the function derivable from the background-theory axioms that represents the variable of interest. The gray manifold represents solutions of the discovered model $f$. The double arrows indicate the distances $\beta(f)$ and $\varepsilon(f)$.

## Interplay between data and theory in AI-Descartes

SR is typically solved with genetic programming (GP)[1–3, 16], however methods based on mixed-integer nonlinear programming (MINLP) have recently been proposed[17–19]. In this work, we develop a new MINLP-based SR solver (described in the Supplementary Information). The input consists of a subset of the operators $\{+, -, \times, \div, \sqrt{}, \log, \exp\}$, an upper bound on expression complexity, and an upper bound on the number of constants used that do not equal 1. Given a dataset, the system formulates multiple MINLP instances to find an expression that minimizes the least-square error. Each instance is solved approximately, subject to a time limit. Both linear and nonlinear constraints can be imposed. In particular, dimensional consistency is imposed when physical dimensions of variables are available.

We use KeYmaera X[20] as a reasoning tool; it is an ATP for hybrid systems and combines different types of reasoning: deductive, real-algebraic, and computer-algebraic reasoning. We also use Mathematica[21] for certain types of analysis of symbolic expressions. While a formula found by any grammar-based system (such as an SR system) is syntactically correct, it may contradict the axioms of the theory or not be derivable from them. In some cases, a formula may not be derivable as the theory may not have enough axioms; the formula may be provable under an extended axiom set or an alternative one (e.g., using a relativistic set of axioms rather than a "Newtonian" one).

An overview of our system seen as a discovery cycle is shown in Fig. 2. Our discovery cycle is inspired by Descartes who advanced the scientific method and emphasized the role that logical deduction, and not empirical evidence alone, plays in forming and validating scientific discoveries. Our present approach differs from implementations of the scientific method that obtain hypotheses from theory and then check them against data; instead we obtain hypotheses from data and assess them against theory. A more detailed schematic of the system is depicted in Fig. 3, where the colored components correspond to the system we present in this work, and the gray components refer to standard techniques for scientific discovery that we have not yet integrated into our current implementation.

## Experimental validation

We tested the different capabilities of our system on three problems (more details in the Methods section). First, we considered the problem of deriving Kepler's third law of planetary motion, providing reasoning-based measures to analyze the quality and generalizablity of the generated formulae. Extracting this law from experimental data is challenging, especially when the masses involved are of very different magnitudes. This is the case for the solar system, where the solar mass is much larger than the planetary masses. The reasoning module helps in choosing between different candidate formulae and identifying the one that generalizes well: using our data and theory integration we were able to re-discover Kepler's third law. We then considered Einstein's time-dilation formula. Although we did not recover this formula from data, we used the reasoning module to identify the formula that generalizes best. Moreover, analyzing the reasoning errors with two different sets of axioms (one with "Newtonian" assumptions and one relativistic), we were able to identify the theory that better explains the phenomenon. Finally, we considered Langmuir's adsorption equation, whose background theory contains material-dependent coefficients. By relating these coefficients to the ones in the SR-generated models via existential quantification, we were able to logically prove one of the extracted formulae.

## Discussion

We have demonstrated the value of combining logical reasoning with symbolic regression in obtaining meaningful symbolic models of physical phenomena, in the sense that they are consistent with background theory and generalize well in a domain that is significantly larger than the experimental data. The synthesis of regression and reasoning yields better models than can be obtained by SR or logical reasoning alone.

Improvements or replacements of individual system components and introduction of new modules such as abductive reasoning or experimental design[22] (not described in this work for the sake of brevity) would extend the capabilities of the overall system. A deeper integration of reasoning and regression can help synthesize models that are both data driven and based on first principles, and lead to a revolution in the scientific discovery process. The discovery of models

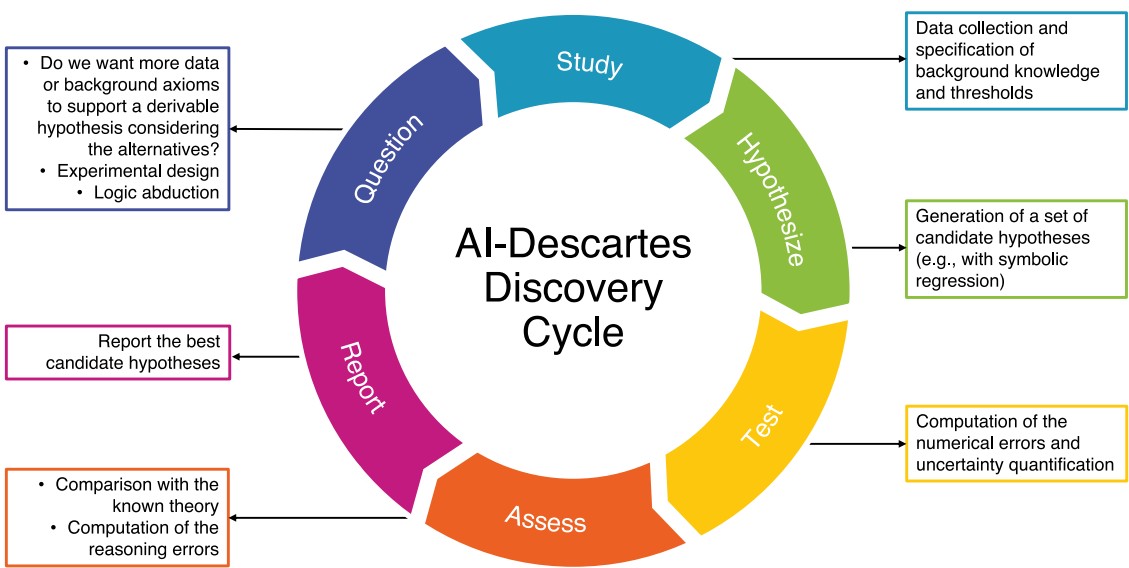

**Fig. 2 | An interpretation of the scientific method as implemented by our system.** The colors match the respective components of the system in Fig. 3.

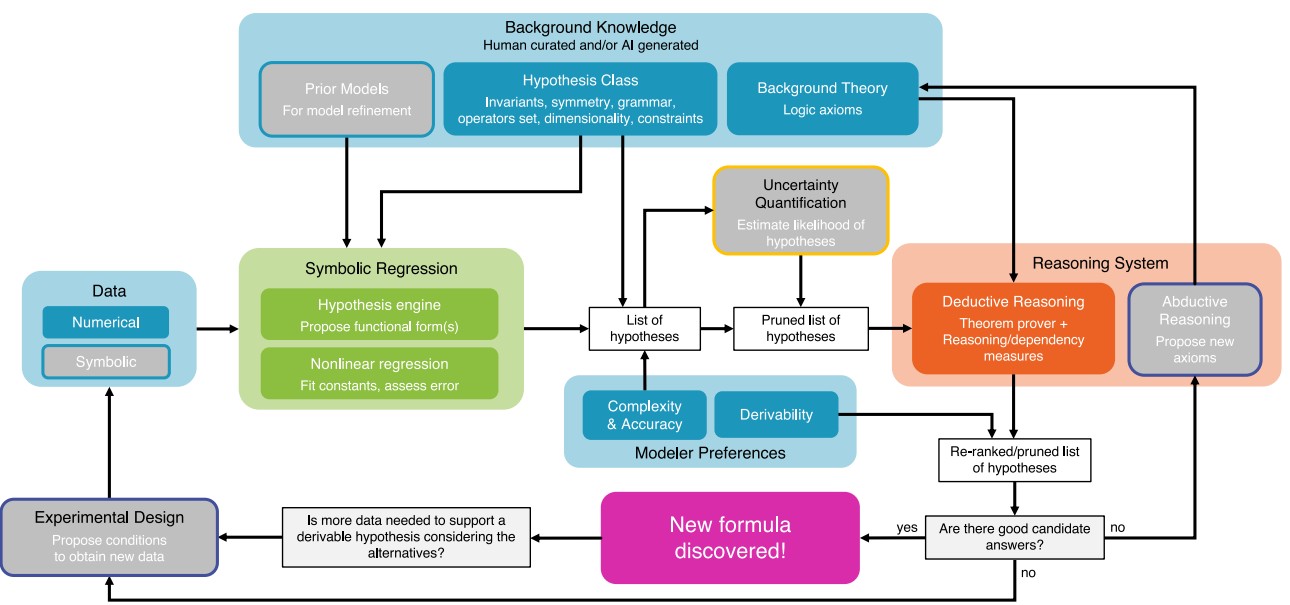

**Fig. 3 | System overview.** Colored components correspond to our system, and gray components indicate standard techniques for scientific discovery (human-driven or artificial) that have not been integrated into the current system. The colors match the respective components of the discovery cycle of Fig. 2. The present system generates hypotheses from data using symbolic regression, which are posed as conjectures to an automated deductive reasoning system, which proves or disproves them based on background theory or provides reasoning-based quality measures.

that are consistent with prior knowledge will accelerate scientific discovery, and enable going beyond existing discovery paradigms.

## Methods

We next describe in detail the methodologies used to address the three problems studied to validate our method: Kepler's third law of planetary motion, relativistic time dilation, and Langmuir's adsorption equation.

### Kepler's third law of planetary motion

Kepler's law relates the distance $d$ between two bodies (e.g., the sun and a planet in the solar system) and their orbital periods. It can be expressed as

$$p = \sqrt{\frac{4\pi^2 d^3}{G(m_1 + m_2)}}, \tag{1}$$

where $p$ is the period, $G$ is the gravitational constant, and $m_1$ and $m_2$ are the two masses. It can be derived using the following axioms of the background theory $\mathcal{B}$, describing the center of mass (axiom K1), the distance between bodies (axiom K2), the gravitational force (axiom K3), the centrifugal force (axiom K4), the force balance (axiom K5), and the period (axiom K6):

| K1. | Center of mass | $m_1 d_1 = m_2 d_2$ |
| K2. | Distance between bodies | $d = d_1 + d_2$ |
| K3. | Gravitational force | $F_g = \frac{G m_1 m_2}{d^2}$ |
| K4. | Centrifugal force | $F_c = m_2 d_2 w^2$ |
| K5. | Force balance | $F_g = F_c$ |
| K6. | Period definition | $p = \frac{2\pi}{w}$ |
| K7. | Positivity constraints | $m_1 > 0, m_2 > 0, p > 0, d_1 > 0, d_2 > 0.$ |

$$\tag{2}$$

We consider three real-world datasets: planets of the solar system (from the NASA Planetary Fact Sheet[23]), the solar-system planets along with exoplanets from Trappist-1 and the GJ 667 system (from the NASA exoplanet archive[24]), and binary stars[25]. These datasets contain measurements of pairs of masses (a sun and a planet for the first two, and two suns for the third), the distance between them, and the orbital period of the planet around its sun in the first two datasets or the orbital period around the common center of mass in the third dataset. The data we use is given in the Supplementary Information. Note that the dataset does not contain measurements for a number of variables in the axiom system, such as $d_1, d_2, F_g$, etc.

The goal is to recover Kepler's third law (Eq. (1)) from the data, that is, to obtain $p$ as the above-stated function of $d$, $m_1$ and $m_2$.

The SR module takes as input the set of operators $\{+, -, \times, \div, \sqrt{}\}$ and outputs a set of candidate formulas. None of the formulae obtained via SR are derivable, though some are close approximations to derivable formulae. We evaluate the quality of these formulae by writing a logic program for calculating the error $\beta$ of a formula with respect to a derivable formula. We use three measures, defined below, to assess the correctness of a data-driven formula from a reasoning viewpoint: the *pointwise reasoning error*, the *generalization reasoning error*, and *variable dependence*.

**Pointwise reasoning error.** The key idea is to compute a distance between a formula generated from the numerical data and some derivable formula that is implicitly defined by the axiom set. The distance is measured by the $l_2$ or $l_\infty$ norm applied to the differences between the values of the numerically-derived formula and a derivable formula at the points in the dataset. This definition can be extended to other norms.

We compute the relative error of numerically derived formula $f(\mathbf{x})$ applied to the $m$ data points $\mathbf{X}^i$ ($i = 1, ..., m$) with respect to $f_\mathcal{B}(\mathbf{x})$, derivable from the axioms via the following expressions:

$$\beta_2^r = \sqrt{\sum_{i=1}^{m} \left(\frac{f(\mathbf{X}^i) - f_\mathcal{B}(\mathbf{X}^i)}{f_\mathcal{B}(\mathbf{X}^i)}\right)^2} \quad \text{and} \quad \beta_\infty^r = \max_{1 \leq i \leq m} \left\{\frac{|f(\mathbf{X}^i) - f_\mathcal{B}(\mathbf{X}^i)|}{|f_\mathcal{B}(\mathbf{X}^i)|}\right\} \tag{3}$$

where $f_\mathcal{B}(\mathbf{X}^i)$ denotes a derivable formula for the variable of interest $y$ evaluated at the data point $\mathbf{X}^i$.

The KeYmaera formulation of these two measures for the first formula of Table 1 can be found in the Supplementary Information. Absolute-error variants of the first and second expressions in Eq. (3)

**Table 1 | Error values of candidate solutions for the Kepler dataset**

| 1<br>Dataset | 2<br>Candidate formula<br>$p =$ | 3<br>numerical error<br>$\varepsilon_2^r$ | 4<br><br>$\varepsilon_\infty^r$ | 5<br>point. reas. err.<br>$\beta_2^r$ | 6<br><br>$\beta_\infty^r$ | 7<br>gen. reas. error<br>$\beta_{\infty,S}^r$ | 8<br>dependencies<br>$m_1$ | 9<br><br>$m_2$ | 10<br><br>$d$ |
|---|---|---|---|---|---|---|---|---|---|
| Solar | $\sqrt{0.1319 d^3}$ | 0.0129 | 0.0064 | 0.0146 | 0.0052 | 0.0052 | 0 | 0 | 1 |
| | $\sqrt{0.1316(d^3 + d)}$ | 1.9348 | 1.7498 | 1.9385 | 1.7533 | 1.7559 | 0 | 0 | 0 |
| | $(0.03765 d^3 + d^2)/(2 + d)$ | 0.3102 | 0.2766 | 0.3095 | 0.2758 | 0.2758 | 0 | 0 | 0 |
| Exoplanet | $\sqrt{0.1319 d^3/m_1}$ | 0.0845 | 0.0819 | 0.0231 | 0.0052 | 0.0052 | 0 | 0 | 1 |
| | $\sqrt{m_1^2 m_2^3/d + 0.1319\, d^3/m_1}$ | 0.1988 | 0.1636 | 0.1320 | 0.1097 | >550 | 0 | 0 | 0 |
| | $\sqrt{(1 - 0.7362 m_1)d^3/2}$ | 1.2246 | 0.4697 | 1.2418 | 0.4686 | 0.4686 | 0 | 0 | 1 |
| Binary stars | $1/(d^2 m_1^2) + 1/(d m_2^2) - m_1^3 m_2^2 + \sqrt{0.4787 d^3/m_2 + d^2 m_2^2}$ | 0.0023 | 0.0015 | 0.0059 | 0.0050 | Timeout | 0 | 0 | 0 |
| | $(\sqrt{d^3} + m_1^3 m_2/\sqrt{d})/\sqrt{m_1 + m_2}$ | 0.0032 | 0.0031 | 0.0038 | 0.0031 | Timeout | 0 | 0 | 0 |
| | $\sqrt{d^3/(0.9967 m_1 + m_2)}$ | 0.0058 | 0.0053 | 0.0014 | 0.0008 | 0.0020 | 1 | 1 | 1 |

Numerical error values, pointwise reasoning error values, and generalization error values are shown. We also give an analysis of the variable dependence of candidate solutions. For simplicity of notation, in the table we use the variables $d$, $m_1$, $m_2$ and $p$, while referring to the scaled counterparts. We assume that all the errors are relative.

are denoted by $\beta_2^a, \beta_\infty^a$, respectively. The numerical (data) error measures $\varepsilon_2^r$ and $\varepsilon_\infty^r$ are defined by replacing $f_\mathcal{B}(\mathbf{X}^i)$ by $Y_i$ in Eq. (3). Analogous to $\beta_2^a$ and $\beta_\infty^a$, we also define absolute-numerical-error measures $\varepsilon_2^a$ and $\varepsilon_\infty^a$.

Table 1 reports in columns 5 and 6 the values of $\beta_2^r$ and $\beta_\infty^r$, respectively. It also reports the relative numerical errors $\varepsilon_2^r$ and $\varepsilon_\infty^r$ in columns 3 and 4, measured by the $l_2$ and $l_\infty$ norms, respectively, for the candidate expressions given in column 2 when evaluated on the points in the dataset. We minimize the absolute $l_2$ error $\varepsilon_2^a$ (and not the relative error $\varepsilon_2^r$), when obtaining candidate expressions via symbolic regression.

The pointwise reasoning errors $\beta_2$ and $\beta_\infty$ are not very informative if SR yields a low-error candidate expression (measured with respect to the data), and the data itself satisfies the background theory up to a small error, which indeed is the case with the data we use; the reasoning errors and numerical errors are very similar.

**Generalization reasoning error.** Even when one can find a function that fits given data points well, it is challenging to obtain a function that generalizes well, that is, one which yields good results at points of the domain not equal to the data points. Let $\beta_{\infty,S}^r$ be calculated for a candidate formula $f(\mathbf{x})$ over a domain $S$ that is not equal to the original set of data points as follows:

$$\beta_{\infty,S}^r = \max_{\mathbf{x} \in S} \left\{ \frac{|f(\mathbf{x}) - f_\mathcal{B}(\mathbf{x})|}{|f_\mathcal{B}(\mathbf{x})|} \right\}, \tag{4}$$

where we consider the relative error and, as before, the function $f_\mathcal{B}(\mathbf{x})$ is not known, but is implicitly defined by the axioms in the background theory. We call this measure the *relative generalization reasoning error*. If we do not divide by $f_\mathcal{B}(\mathbf{x})$ in the above expression, we get the corresponding *absolute* version $\beta_{\infty,S}^a$. For the Kepler dataset, we let $S$ be the smallest multi-dimensional interval (or Cartesian product of intervals on the real line) containing all data points. In column 7 of Table 1, we show the relative generalization reasoning error $\beta_{\infty,S}^r$ on the Kepler datasets with $S$ defined as above. If this error is roughly the same as $\beta_\infty^r$ the pointwise relative reasoning error for $l_\infty$ (e.g., for the solar system dataset) then the formula extracted from the numerical data is as accurate at points in $S$ as it is at the original data points.

**Variable dependence.** In order to check if the functional dependence of a candidate formula on a specific variable is accurate, we compute

the generalization error over a domain $S'$ where the domain of this variable is extended by an order of magnitude beyond the smallest interval containing the values of the variable in the dataset. Thus we can check whether there exist special conditions under which the formula does not hold. We modify the endpoints of an interval by one order of magnitude, one variable at a time. If we notice an increase in the generalization reasoning error while modifying intervals for one variable, we deem the candidate formula as missing a dependency on that variable. A missing dependency might occur, for example, because the exponent for a variable is incorrect, or that variable is not considered at all when it should be. One can get further insight into the type of dependency by analyzing how the error varies (e.g., linearly or exponentially). Table 1 provides, in columns 8–10, results regarding the candidate formulae for Kepler's third law. For each formula, the dependencies on $m_1$, $m_2$, and $d$ are indicated by 1 or 0 (for correct or incorrect dependency). For example, the candidate formula $p = \sqrt{0.1319 d^3}$ for the solar system does not depend on either mass, and the dependency analysis suggests that the formula approximates well the phenomenon in the solar system, but not for larger masses.

The best formula for the binary-star dataset, $\sqrt{d^3/(0.9967 m_1 + m_2)}$, has no missing dependency (all ones in columns 8–10), that is, it generalizes well; increasing the domain along any variable does not increase the generalized reasoning error.

Figure 4 provides a visualization of the two errors $\varepsilon_2^r$ and $\beta_2^r$ for the first three functions of Table 1 (solar-system dataset) and the ground truth $f^*$.

**Relativistic time dilation**
Einstein's theory of relativity postulates that the speed of light is constant, and implies that two observers in relative motion to each other will experience time differently and observe different clock frequencies. The frequency $f$ for a clock moving at speed $v$ is related to the frequency $f_0$ of a stationary clock by the formula

$$\frac{f - f_0}{f_0} = \sqrt{1 - \frac{v^2}{c^2}} - 1, \tag{5}$$

where $c$ is the speed of light. This formula was recently confirmed experimentally by Chou et al.[26] using high precision atomic clocks. We test our system on the experimental data reported by Chou et al.[26] which consists of measurements of $v$ and associated values of

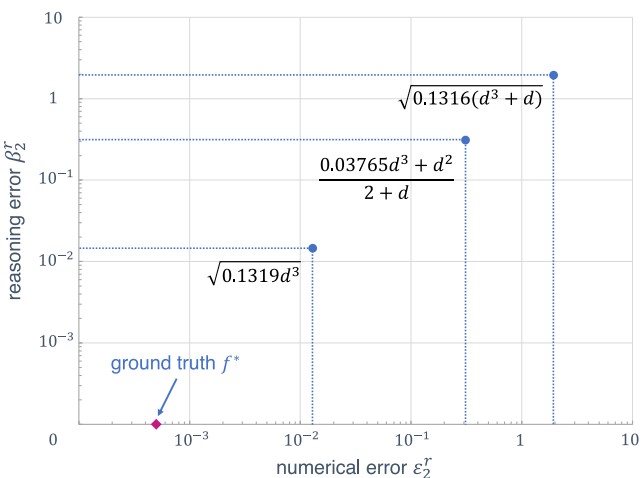

**Fig. 4 | Depiction of symbolic models for Kepler's third law of planetary motion giving the orbital period of a planet in the solar system.** The models produced by our SR system are represented by points $(\varepsilon, \beta)$, where $\varepsilon$ represents distance to data, and $\beta$ represents distance to background theory. Both distances are computed with an appropriate norm on the scaled data.

$(f - f_0)/f_0$, reproduced in the Supplementary Information. We take the axioms for derivation of the time dilation formula from the work of Behroozi[27] and Smith[28]. These are also listed in the Supplementary Information and involve variables that are not present in the experimental data.

In Table 2 we give some functions obtained by our SR module (using $\{+, -, \times, \div, \sqrt{\ }\}$ as the set of input operators) along with the numerical errors of the associated functions and generalization reasoning errors. The sixth column gives the set $S$ as an interval for $v$ for which our reasoning module can verify that the absolute generalization reasoning error of the function in the first column is at most 1. The last column gives the interval for $v$ for which we can verify a relative generalization reasoning error of at most 2%. Even though the last function has low relative error according to this metric, it can be ruled out as a reasonable candidate if one assumes the target function should be continuous (it has a singularity at $v=1$). Thus, even though we cannot obtain the original function, we obtain another which generalizes well, as it yields excellent predictions for a very large range of velocities.

In this case, our system can also help rule out alternative axioms. Consider replacing the axiom that the speed of light is a constant value $c$ by a "Newtonian" assumption that light behaves like other mechanical objects: if emitted from an object with velocity $v$ in a direction perpendicular to the direction of motion of the object, it has velocity $\sqrt{v^2 + c^2}$. Replacing $c$ by $\sqrt{v^2 + c^2}$ (in axiom R2 in the Supplementary Information to obtain R2') produces a self-consistent axiom system (as confirmed by the theorem prover), albeit one leading to no time dilation. Our reasoning module concludes that none of the functions in Table 2 is compatible with this updated axiom system: the absolute generalization reasoning error is greater than 1 even on the dataset domain, as well as the pointwise reasoning error. Consequently, the data is used indirectly to discriminate between axiom systems relevant for the phenomenon under study; SR poses only accurate formulae as conjectures.

**Langmuir's adsorption equation**
The Langmuir adsorption equation (Nobel Prize in Chemistry, 1932)[29] describes a chemical process in which gas molecules contact a surface, and relates the loading $q$ on the surface to the pressure $p$ of the gas:

$$q = \frac{q_{max} K_a p}{1 + K_a p} . \tag{6}$$

The constants $q_{max}$ and $K_a$ characterize the maximum loading and the adsorption strength, respectively. A similar model for a material with two types of adsorption sites yields:

$$q = \frac{q_{max,1} K_{a,1} p}{1 + K_{a,1} p} + \frac{q_{max,2} K_{a,2} p}{1 + K_{a,2} p} , \tag{7}$$

with parameters for maximum loading and adsorption strength on each type of site. The parameters in Eqs. (6) and (7) fit experimental data using linear or nonlinear regression, and depend on the material, gas, and temperature.

We used data from Langmuir's 1918 publication[29] for methane adsorption on mica at a temperature of 90 K, and also data from the work of Sun et al.[30] (Table 1) for isobutane adsorption on silicalite at a temperature of 277 K. In both cases, observed values of $q$ are given for specific values of $p$; the goal is to express $q$ as a function of $p$. We give the SR module the operators $\{+, -, \times, \div\}$, and obtain the best fitting functions with two and four constants. The code ran for 20 minutes on 45 cores, and seven of these functions are displayed for each dataset.

To encode the background theory, following Langmuir's original theory[29], we elicited the following set $\mathcal{A}$ of axioms:

| | | |
|---|---|---|
| L1. | Site balance | $S_0 = S + S_a$ |
| L2. | Adsorption rate model | $r_{ads} = k_{ads} p S$ |
| L3. | Desorption rate model | $r_{des} = k_{des} S_a$ |
| L4. | Equilibrium assumption | $r_{ads} = r_{des}$ |
| L5. | Mass balance on $q$ | $q = S_a$. |

(8)

Here, $S_0$ is the total number of sites, of which $S$ are unoccupied and $S_a$ are occupied (L1). The adsorption rate $r_{ads}$ is proportional to the pressure $p$ and the number of unoccupied sites (L2). The desorption rate $r_{des}$ is proportional to the number of occupied sites (L3). At equilibrium, $r_{ads} = r_{des}$ (L4), and the total amount adsorbed, $q$, is the number of occupied sites (L5) because the model assumes each site adsorbs at most one molecule. Langmuir solved these equations to obtain

$$q = \frac{S_0 (k_{ads}/k_{des}) p}{1 + (k_{ads}/k_{des}) p} , \tag{9}$$

which corresponds to Eq. (6), where $q_{max} = S_0$ and $K_a = k_{ads}/k_{des}$. An axiomatic formulation for the multi-site Langmuir expression is described in the Supplementary Information. Additionally, constants and variables are constrained to be positive (e.g., $S_0 > 0$, $S > 0$, and $S_a > 0$) or non-negative (e.g., $q \geq 0$).

The logic formulation to prove is:

$$(\mathcal{C} \wedge \mathcal{A}) \rightarrow f , \tag{10}$$

where $\mathcal{C}$ is the conjunction of the non-negativity constraints, $\mathcal{A}$ is a conjunction of the axioms, the union of $\mathcal{C}$ and $\mathcal{A}$ constitutes the background theory $\mathcal{B}$, and $f$ is the formula we wish to prove.

SR can only generate numerical expressions involving the (dependent and independent) variables occurring in the input data, with certain values for constants; for example, the expression $f = p/(0.709p + 0.157)$. The expressions built from variables and constants from the background theory, such as Eq. (9), involve the constants (in their symbolic form) explicitly: for example, $k_{ads}$ and $k_{des}$ appear explicitly in Eq. (9) while SR only generates a numerical instance of the ratio of these constants. Thus, we cannot use Formula (10) directly to prove formulae generated from SR. Instead, we replace each numerical constant of the formula by a logic variable $c_i$: for example, the formula $f = p/(0.709p + 0.157)$ is replaced by $f' = p/(c_1 p + c_2)$, introducing two new variables $c_1$ and $c_2$. We then quantify the new variables

**Table 2 | Candidate functions derived from time dilation data, and associated error values**

| Candidate formula | Numerical Error Absolute | | Numerical Error Relative | | S s.t. Absolute Gen. Reas. Error | S s.t. Relative Gen. Reas. Error |
|---|---|---|---|---|---|---|
| $f =$ | $\varepsilon_2^a$ | $\varepsilon_\infty^a$ | $\varepsilon_2^r$ | $\varepsilon_\infty^r$ | $\beta_{\infty,\mathbf{S}}^a \leq 1$ | $\beta_{\infty,\mathbf{S}}^r \leq .02$ |
| $-0.00563v^2$ | 0.3822 | 0.3067 | 1.0811 | 0.0018 | $37 \leq v \leq 115$ | $37 \leq v \leq 10^8$ |
| $\frac{v}{1+0.00689v} - v$ | 0.3152 | 0.2097 | 1.0125 | 0.0069 | $37 \leq v \leq 49$ | $37 \leq v \leq 38$ |
| $-0.00537\frac{v^2\sqrt{v+v^2}}{(v-1)}$ | 0.3027 | 0.2299 | 1.2544 | 0.0021 | $37 \leq v \leq 98$ | $37 \leq v \leq 109$ |
| $-0.00545\frac{v^4}{\sqrt{v^2+v^{-2}}(v-1)}$ | 0.3238 | 0.2531 | 1.1308 | 0.0010 | $37 \leq v \leq 126$ | $37 \leq v \leq 10^7$ |

The values of $v$ are defined in $m/s$.

existentially, and define a new set of non-negativity constraints $\mathcal{C}'$. In the example above we will have $\mathcal{C}' = c_1 > 0 \,\wedge\, c_2 > 0$.

The final formulation to prove is:

$$\exists c_1 \cdots \exists c_n (\mathcal{C} \wedge \mathcal{A}) \rightarrow (f' \wedge \mathcal{C}'). \tag{11}$$

For example, $f' = p/(c_1 p + c_2)$ is proved true if the reasoner can prove that there exist values of $c_1$ and $c_2$ such that $f'$ satisfies the background theory $\mathcal{A}$ and the constraints $\mathcal{C}$. Here $c_1$ and $c_2$ can be functions of constants $k_{ads}$, $k_{des}$, $S_0$, and/or real numbers, but not the variables $q$ and $p$.

We also consider background knowledge in the form of a list of desired properties of the relation between $p$ and $q$, which helps trim the set of candidate formulae. Thus, we define a collection $\mathcal{K}$ of constraints on $f$, where $q = f(p)$, enforcing monotonicity or certain types of limiting behavior (see Supplementary Information). We use Mathematica[21] to verify that a candidate function satisfies the constraints in $\mathcal{K}$.

In Table 3, column 1 gives the data source, and column 2 gives the "hyperparameters" used in our SR experiments: we allow either two or four constants in the derived expressions. Furthermore, as the first constraint C1 from $\mathcal{K}$ can be modeled by simply adding the data point $p = q = 0$, we also experiment with an "extra point".

Column 3 displays a derived expression, while columns 4 and 5 give, respectively, the relative numerical errors $\varepsilon_2^r$ and $\varepsilon_\infty^r$. If the expression can be derived from our background theory, then we indicate that in column 6. These results are visualized in Fig. 5. Column 7 indicates the number of constraints from $\mathcal{K}$ that each expression satisfies, verified by Mathematica. Among the top two-constant expressions, $f_1$ fits the data better than $f_2$, which is derivable from the background theory, whereas $f_1$ is not.

When we search for four-constant expressions[29], we get much smaller errors than Eq. (6) or even Eq. (7), but we do not obtain the two-site formula (Eq. (7)) as a candidate expression. For the dataset from Sun et al.[30], $g_2$ has a form equivalent to Langmuir's one-site formula, and $g_5$ and $g_7$ have forms equivalent to Langmuir's two-site formula, with appropriate values of $q_{max,i}$ and $K_{a,i}$ for $i = 1, 2$.

**System limitations and future improvements**

Our results on three problems and associated data are encouraging and provide the foundations of a new approach to automated scientific discovery. However our work is only a first, although crucial, step towards completing the missing links in automating the scientific method.

One limitation of the reasoning component is the assumption of correctness and completeness of the background theory. The incompleteness could be partially solved by the introduction of abductive reasoning[31] (as depicted in Fig. 3). Abduction is a logic technique that aims to find explanations of an (or a set of) observation, given a logical theory. The explanation axioms are produced in a way that satisfy the following: (1) the explanation axioms are consistent with the original logical theory and (2) the observation can be deduced by the new

enhanced theory (the original logical theory combined with the explanation axioms). In our context the logical theory corresponds to the set of background knowledge axioms that describe a scientific phenomenon, the observation is one of the formulas extracted from the numerical data and the explanations are the missing axioms in the incomplete background theory.

However the availability of background theory axioms in machine readable format for physics and other natural sciences is currently limited. Acquiring axioms could potentially be automated (or partially automated) using knowledge extraction techniques. Extraction from technical books or articles that describe a natural science phenomenon can be done by, for example, deep learning methods (e.g. the work of Pfahler and Morik[32], Alexeeva et al.[33], or Wang and Liu[34]) both from NL plain text or semi-structured text such as LateX or HTML. Despite the recent advancements in this research field, the quality of the existing tools remains quite inadequate with respect to the scope of our system.

Another limitation of our system, that heavily depends on the tools used, is the scaling behavior. Excessive computational complexity is a major challenge for automated theorem provers (ATPs): for certain types of logic (including the one that we use), proving a conjecture is undecidable. Deriving models from a logical theory using formal reasoning tools is even more difficult when using complex arithmetic and calculus operators. Moreover, the run-time variance of a theorem prover is very large: the system can at times solve some "large" problems while having difficulties with some "smaller" problems. Recent developments in the neuro-symbolic area use deep-learning techniques to enhance standard theorem provers (e.g., see Crouse et al.[8]). We are still at the early stages of this research and there is still a lot that can be done. We envision that the performance and capability (in terms of speed and expressivity) of theorem provers will improve with time. Symbolic regression tools, including the one based on solving mixed-integer nonlinear programs (MINLP) that we developed, often take an excessive amount of time to explore the space of possible symbolic expressions and find one that has low error and expression complexity, especially with noisy data. In practice, the worst-case solution time for MINLP solvers (including BARON) grows exponentially with input data encoding size (additional details in the Supplementary Information). However, MINLP solver performance and genetic programming based symbolic regression solvers are active areas of research.

Our proposed system could benefit from other improvements in individual components (especially in the functionality available). For example, Keymaera only supports differential equations in time and not in other variables and does not support higher order logic; BARON cannot handle differential equations.

Beyond improving individual components, our system can be improved by introducing techniques such as experimental design (not described in this work but envisioned in Fig. 3). A fundamental question in the holistic view of the discovery process is what data should be collected to give us maximum information regarding the underlying model. The goal of optimal experimental design (OED) is to find an

**Table 3 | Results on two datasets for the Langmuir problem**

| 1 Data | 2 Condition | 3 Candidate formula $q =$ | 4 Numerical Error $\varepsilon_2^r$ | 5 $\varepsilon_\infty^r$ | 6 KeYmaera provability | 7 $\mathcal{K}$ constr. |
|---|---|---|---|---|---|---|
| Langmuir[29] (Table IX) | 2 const. | $f_1$: $(p^2 + 2p - 1)/(0.00888p^2 + 0.118p)$ | 0.0631 | 0.0486 | Timeout | 2/5 |
| | | $f_2$: $p/(0.00927p + 0.0759)$ * | 0.1799 | 0.1258 | Yes | 5/5 |
| | 4 const. | $f_3$: $(p^2 - 10.5p - 15)/(0.00892p^2 - 1.23)$ | 0.0443 | 0.0295 | Timeout | 2/5 |
| | | $f_4$: $(8.86p + 13.9)/(0.0787p + 1)$ | 0.0658 | 0.0465 | No | 4/5 |
| | | $f_5$: $p^2/(0.00895p^2 + 0.0934p - 0.0860)$ | 0.0759 | 0.0496 | No | 2/5 |
| | 4 const. extra-point | $f_6$: $(p^2 + p)/(0.00890p^2 + 0.106p - 0.0311)$ | 0.0683 | 0.0470 | Timeout | 2/5 |
| | | $f_7$: $(112p^2 - p)/(p^2 + 10.4p - 9.66)$ | 0.0771 | 0.0532 | Timeout | 3/5 |
| Sun et al.[30] (Table 1) | 2 const. | $g_1$: $(p + 3)/(0.584p + 4.01)$ | 0.1625 | 0.1007 | No | 4/5 |
| | | $g_2$: $p/(0.709p + 0.157)$ | 0.9680 | 0.5120 | Yes | 5/5 |
| | 4 const. | $g_3$: $(0.0298p^2 + 1)/(0.0185p^2 + 1.16) - 0.000905/p^2$ | 0.1053 | 0.0538 | Timeout | 2/5 |
| | | $g_4$: $1/(p^2 + 1) + (2.53p - 1)/(1.54p + 2.77)$ | 0.1300 | 0.0725 | Timeout | 3/5 |
| | 4 const. extra-point | $g_5$: $(1.74p^2 + 7.61p)/(p^2 + 9.29p + 0.129)$ | 0.1119 | 0.0996 | Timeout | 5/5 |
| | | $g_6$: $(0.226p^2 + 0.762p - 7.62 \cdot 10^{-4})/(0.131p^2 + p)$ | 0.1540 | 0.0935 | Timeout | 2/5 |
| | | $g_7$: $(4.78p^2 + 26.6p)/(2.71p^2 + 30.4p + 1)$ | 0.1239 | 0.1364 | Timeout | 5/5 |

(* This expression is also generated when using four constants, and also when the extra point $(0, 0)$ is added).

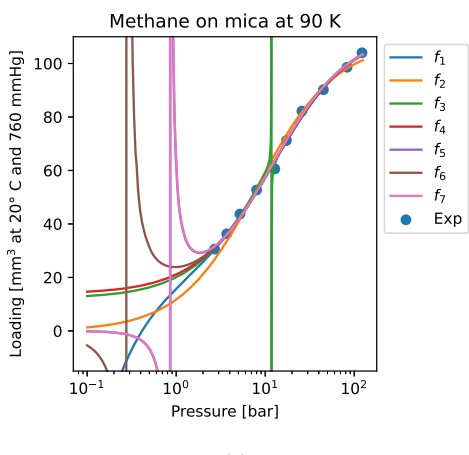
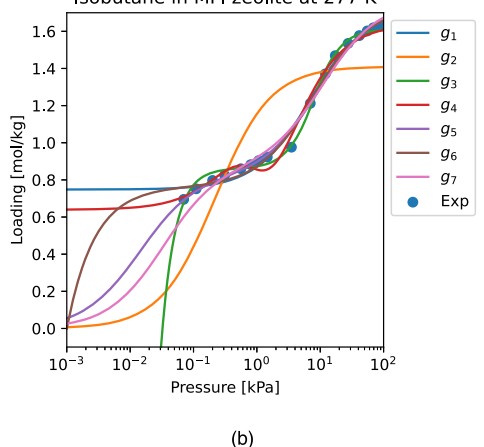

**Fig. 5 | Symbolic regression solutions to two adsorption datasets.** Fig. 5a refers to the methane adsorption on mica at a temperature of 90 K, while Fig. 5b refers to the isobutane adsorption on silicalite at a temperature of 277 K. $f_2$ and $g_2$ are equivalent to the single-site Langmuir equation; $g_5$ and $g_7$ are equivalent to the two-site Langmuir equation.

optimal sequence of data acquisition steps such that the uncertainty associated with the inferred parameters, or some predicted quantity derived from them, is minimized with respect to a statistical or information theoretic criterion. In many realistic settings, experimentation may be restricted or costly, providing limited support for any given hypothesis as to the underlying functional form. It is therefore critical at times to incorporate an effective OED framework. In the context of model discovery, a large body of work addresses the question of experimental design for predetermined functional forms, and another body of research addresses the selection of a model (functional form) out of a set of candidates. A framework that can deal with both the functional form and the continuous set of parameters that define the model behavior is obviously desirable[22]; one that consistently accounts for logical derivability or knowledge-oriented considerations[35] would be even better.

## Data availability
The data used in this study are available in the AI-Descartes GitHub repository[36]: https://github.com/IBM/AI-Descartes.

## Code availability
The code used for this work can be found, freely available, at the AI-Descartes GitHub repository[36]: https://github.com/IBM/AI-Descartes.

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

## Acknowledgements

We thank J. Ilja Siepmann for initially suggesting adsorption as a problem for symbolic regression. We thank James Chin-wen Chou for providing the atomic clock data. Funding: This work was supported in part by the Defense Advanced Research Projects Agency (DARPA) (PA-18-02-02). The U.S. Government is authorized to reproduce and distribute reprints for Governmental purposes notwithstanding any copyright notation thereon. T.R.J. was supported by the U.S. Department of Energy (DOE), Office of Basic Energy Sciences, Division of Chemical Sciences, Geosciences and Biosciences (DE-FG02-17ER16362), as well as startup funding from the University of Maryland, Baltimore County. T.R.J. also gratefully acknowledges the University of Minnesota Institute for Mathematics and its Applications (IMA).

## Author contributions

C.C. conceptualized the overarching derivable symbolic discovery architecture, designed the project, designed and implemented the reasoning module, designed and discussed the experiments, performed the experiments for the reasoning module and for the comparison with the state of the art, analyzed and formatted the data, formalized the scientific theories, formatted code and data for the release, wrote and edited the manuscript, and designed the figures. S.D. conceptualized the overarching derivable symbolic discovery architecture, designed the project, designed the SR module architecture, designed and discussed the experiments, performed the experiments for the SR module, analyzed and formatted the data, and wrote and edited the manuscript. V.A. designed and implemented the SR module. T.R.J. identified target problems and experimental datasets, formalized the scientific theories, discussed the experiments, designed the figures, and wrote and edited the manuscript. J.G. prepared the data, executed the computational experiments for the SR module and for the comparison with the state of the art, formatted code and data for the release, and edited the manuscript. K.C. discussed and designed the overarching project, discussed the experiments, and edited and revised the manuscript. N.M. discussed and designed the overarching project, discussed the experiments, provided conceptual advice, and edited the manuscript. B.E.K. designed figure 1, discussed the reasoning measures, and edited the manuscript. L.H. conceptualized the overarching derivable symbolic discovery architecture, designed the project, designed the experiments, analyzed the results per validation of the framework, and wrote and edited the manuscript.

## Competing interests

The authors declare no competing interests.
