## [Peer Review File · Nature Communications]

REVIEWER COMMENTS

Reviewer #1 (Remarks to the Author):

AI Descartes presents an interesting challenge of discovering proper equations from data. While current version describe basic ideas and outcomes, the paper can be improved significantly.

(1) Defining "Background knowledge" is a critical part of the process in AI Descartes to function as it defines domain specific axioms. Is there any chance even such domain knowledge can be delivered automatically? I understand this is outside of the scope of the paper. However, such approach could be possible using deep learning based system with explanation.

(2) Hypothesis Space. Very curious to understand complexity of hypothesis space. Provided a set of axioms in Background knowledge, complexity (a size of the search space) can be shown with certain conditions such as numbers of axioms or symbolic elements of equations to be used. It would be useful for readers to understand what is the size of problems the system is solving.

(3) Figure 1 is hard to understand. description of mathematical structure of the problem is well written but Figure 1 may not be the best visual description. for example, background theory looks like continuous space where the task is to find parametric optima. the background theory is not merely parametric which is different from impression I got from the Figure 1. etc. etc.

I can not give you solution for this, but hope to have better visualization.

(4) Conversion process?

another point that I wish to know is what is the process of discovery. does it generate a large equations, I guess, and how quickly does it converge and does it trapped into local minima ? if so, how does it overcome?

for example, figure 1 in supplementary material (3/8) indicate points in numerical error / reasoning error phase space, but over the course of inference what are convergence or initial hypothesis distribution moved in the phase space to get these solutions?

(5) AAI for Scientific Discovery is a growing field and in its infancy. the best papers at this moment is to uncover outcome and issues in methods rather than simply demonstrating what can be done.

in this regards, I would recommend authors to uncover problems the system is facing, complexity of the problems, and what are the options that can be taken to overcome these issues.

Reviewer #2 (Remarks to the Author):

Paper Summary

The paper proposes an approach for data-driven discovery of mathematical expressions describing a physical system using (a) symbolic regression to discover candidate expressions and (b) then using an automated theorem prover to assess the quality of discovered expressions with respect to a certain set of constraints or axioms.

The authors propose a new MILP solver to achieve (a) and to achieve (b) they use an existing open-source tool. They also propose a metric to evaluate the quality of expressions generated in (a) using (b).

They show experimental results from three different physical systems with varying levels of success and argue that such an approach can be

improved to accelerate scientific discovery.

Novelty

In terms of the idea, the approach of combining (a) symbolic and (b) formal methods for scientific discovery is not novel. In terms of the formalism or tools used, the paper doesn't discuss any significant improvements to the existing approaches used to achieve (a) or (b). As far as I know the idea was first proposed in the logic guided machine learning paper Scott et al. at AAI 2020 and Ashok et al. at AAI 2021, cited by the authors of this paper. However, the authors of this paper do not give enough credit to the Scott et al. or the Ashok et al. papers. In particular, their abstract needs to refer to both Scott et al. and Ashok et al., in order for the paper to be acceptable.

The main novelty of the paper, in my opinion, is the specific approach described by the authors to combine symbolic regression and automated theorem provers (specifically, MILP) to evaluate the quality of expressions generated the symbolic regression engine. Thus, the paper can perhaps be best thought of as a "toy-model" describing a powerful idea already explored in previous research.

Q1: How is your work conceptually different from the above-mentioned papers?

Q2: Can you go beyond toy problems?

Quality of Writing

The paper is well-written with excellent quality graphical representations describing the overall idea. However, the paper falls short when it comes providing sufficient quantitative evidence (both in terms of the scale of the experiments and the choice of systems used for experimentation) to allow to reader to discern the utility of the proposed approach on realistic examples.

Concerns

The numerical errors reported in the paper are quite large ($\sim 1e-3$). Further, it seems like the more complex the expressions (cf. solar and binary stars in Table 1) lower are the average case numerical and reasoning errors. Is this approach, then, likely to 'over-fit', despite the use of domain knowledge? In particular, how do you decide between two candidate expressions, one with lower reasoning error and higher numerical error vs an expression with higher reasoning error and lower numerical error?

Further, for deriving Kepler's law, the paper uses Newton's law. What happens you remove that axiom? How do your errors change? More generally, what are the minimum number of axioms required to distinguish between the candidate expressions in the experiments reported? This can be a very powerful additional line of research that I think the authors ought to consider.

Potential long-term impact

The idea proposed in the paper could be of significant long-term potential impact; however, as it stands, it is likely not mature enough to contribute to scientific progress. The question then, is whether such progress is possible.

Symbolic regression is still in its early stages of development and it might not be unreasonable to assume they will improve with time and be able to derive increasingly complex expressions with higher numerical accuracy. However, can automated theorem provers however, keep up with the associated complexity?

Most physical phenomena are often described using expressions that are simple closed form algebraic expressions. For example, underlying laws are often described in the language of probabilistic models or partial differential equations. If the idea is to use such a method for scientific discovery, in my opinion, it will necessary for both symbolic regression to be able to derive such expressions and automated theorem provers to be able to reason about them.

Overall comment

The paper proposes a novel idea to combining machine learning and formal reasoning, based on previous work. However, it shows insufficient experimental evidence to convince the reader of its utility for real

scientific problems. I wouldn't be inclined to publish this in Nature in its current form. Here are concrete suggestions for improvement:

- * Cite previous work appropriately and prominently, including in the abstract

- * Discuss limits of the method in detail

- * Provide detailed empirical comparisons with other symbolic regression approaches such as AI Feynman, LGML, LGGGA, Wetzal et al., Eureqa, TuringBot etc.

- * Try adding different axioms to see whether the system can learn a new expression with minimal domain knowledge (where is the boundary for specific cases?)

- * The numerical errors in the work presented are high. Can they be lowered?

- * More experiments need to be done to better understand scalability (in terms of the length and complexity of symbolic expressions learnt)

Reviewer #3 (Remarks to the Author):

Needless to say, the topic covered in the paper is of enormous scientific interest! The paper creates the conditions for stimulating the reading and I also think that it comes at the right time in the overall research on Machine Learning. The examples that are chosen for claiming the generality of the proposed inferential mechanism are quite different, which leads to believe that we are in front of a paper which can offer solid foundations in this field.

Unfortunately, the curious reader is likely to be quite disappointed when trying to disclose and understand the basic ideas. Fig. 3 is nice, yet it is not supported by specific technical descriptions in the "Methods" section. The inferential mechanisms that emerge from data and, especially, from symbolic knowledge are only superficially described. One very much would like to appreciate know symbolic descriptions of the base knowledge activates the inferential process and, particularly, how data promotes different inferential paths.

In the current form, the paper is far from providing a convincing general scheme that can be used to support inference in different research domain. Along with more technical details, the code availability could help to understand the actual value of the paper.

Revisions of: AI Descartes: Combining Data and Theory for Derivable Scientific Discovery

To the reviewers:

We thank you for your careful consideration of our paper. We have done an extensive round of revisions focusing on the major issues that you pointed out.

All the modifications in the paper (main, methods and supplementary material) are marked in blue color.

In what follows, you can find our responses to the comments preceded by >>> and appearing in blue bold font.

The code used for the experiments is now publicly available at <https://github.com/IBM/AI-Descartes>.

Reviewer #1

(Remarks to the Author):

AI Descartes presents an interesting challenge of discovering proper equations from data. While current version describe basic ideas and outcomes, the paper can be improved significantly.

(1) Defining "Background knowledge" is a critical part of the process in AI Descartes to function as it defines domain specific axioms. Is there any chance even such domain knowledge can be delivered automatically? I understand this is outside of the scope of the paper. However, such approach could be possible using deep learning based system with explanation.

>>> The availability of background theory in the form of axioms for physics and other natural sciences is currently very limited. As you mentioned, this could be potentially automated (or partially automated) using knowledge extraction techniques. The extraction from technical books or articles that describe a natural science phenomenon can be done both from NL plain text or semi structured text such as LaTeX or HTML. There are several methods available in the literature, mostly based on deep-learning e.g. Pfahler and Morik "Semantic Search in Millions of Equations" (KDD '20), Alexeeva et al. "MathAlign: Linking Formula Identifiers to their Contextual Natural Language Descriptions" (LREC 2020), and Wang and Liu "Translating Math Formula Images to LaTeX Sequences Using Deep Neural Networks with Sequence-level Training" (IJ DAR-2021).

However, despite all the recent advancement in this research field, the quality of the existing tools remains quite inadequate with respect to the needs of our system. We added a paragraph talking about this in the new section "System limitations and future improvements" at the end of the Methods part of the paper.

(2) Hypothesis Space. Very curious to understand complexity of hypothesis space. Provided a set of axioms in Background knowledge, complexity (a size of the search space) can be shown with certain conditions such as numbers of axioms or symbolic elements of equations to be used. It would be useful for readers to understand what is the size of problems the system is solving.

>>> The "size of the search space" may not be a good measure for the difficulty of a problem. First of all, in first order logic with function symbols, this space is infinite. Second, the "search" is sophisticated to the extent that it sometimes finds results quickly in a large space, but may fail in a smaller space. The running time required to solve MILP and MINLP problems is not as easy to predict as in other computational problems like sorting. Solving different MINLP instances of the same size may require extremely different running times. This behavior also occurs in theorem-proving. Deduction in first order logic with basic arithmetic is an undecidable problem (Gödel's first incompleteness theorem). Therefore, the variance of the run-time of a theorem

prover is very large: the system we use can, on the one hand, solve some “large” problems (because the desired theorems have short proofs that can be discovered relatively quickly), and on the other hand, have difficulties with some smaller problems. Furthermore, it is not clear how the “size” of a theorem-proving problem should be defined. There are several parameters that may affect the size: for example, the number and length of the axioms, the number of distinct predicates, functions and constants, the number of quantifiers (or the number of alternations between existential and universal quantifiers), etc. However, there are several ways to write the same set of axioms: connecting all of them as a single rule (that is the conjunction of all the axioms), or by introducing new auxiliary predicates and variables to split the axioms and thus reducing their length.

(3) Figure 1 is hard to understand. description of mathematical structure of the problem is well written but Figure 1 may not be the best visual description. for example, background theory looks like continuous space where the task is to find parametric optima. the background theory is not merely parametric which is different from impression I got from the Figure 1. etc. etc. I can not give you solution for this, but hope to have better visualization.

>>> We have changed the caption of Figure 1. The two surfaces represent the solutions of the derivable (from the background theory axioms) formula and the formula discovered from data respectively. In this figure we depict Kepler’s problem where both such formulas define a continuous manifold.

(4) Conversion process? another point that I wish to know is what is the process of discovery. does it generate a large equations, I guess, and how quickly does it converge and does it trapped into local minima ? if so, how does it overcome?

>>> We have described in detail how we solve symbolic regression problems from data in the supplementary material. In particular, we explain how we create multiple generalized expression trees, and solve a mixed-integer nonlinear programming (MINLP) problem per “gentree”. We note however that even a single MINLP can take a lot of time, and we explain this in a new section titled “Scaling” in the supplementary material, along with figures showing convergence to the optimal solution with time.

For example, figure 1 in supplementary material (3/8) indicate points in numerical error / reasoning error phase space, but over the course of inference what are convergence or initial hypothesis distribution moved in the phase space to get these solutions?

>>> (We believe the reviewer is pointing to Figure 1 in the methods section; our submission has the main manuscript, a methods part, and supplementary material) We have now added a description of scaling and convergence for our symbolic regression module in the new section titled “Scaling” in the supplementary material. We also added Figures 3 and 4 to illustrate convergence in two instances. However, in our symbolic regression module, we do not have any control over the movement of reasoning error, and only try to reduce numerical error.

(5) AAI for Scientific Discovery is a growing field and in its infancy. the best papers at this moment is to uncover outcome and issues in methods rather than simply demonstrating what can be done.

in this regards, I would recommend authors to uncover problems the system is facing, complexity of the problems, and what are the options that can be taken to overcome these issues.

>>> We added a discussion in the paper regarding the limitations of our proposed system (see the new section “System limitation and future improvements” in the Methods part of the paper).

Reviewer #2

(Remarks to the Author):

Paper Summary

The paper proposes an approach for data-driven discovery of mathematical expressions describing a physical system using (a) symbolic regression to discover candidate expressions and (b) then using an automated theorem prover to assess the quality of discovered expressions with respect to a certain set of constraints or axioms.

The authors propose a new MILP solver to achieve (a) and to achieve (b) they use an existing open-source tool. They also propose a metric to evaluate the quality of expressions generated in (a) using (b).

They show experimental results from three different physical systems with varying levels of success and argue that such an approach can be improved to accelerate scientific discovery.

Novelty

In terms of the idea, the approach of combining (a) symbolic and (b) formal methods for scientific discovery is not novel. In terms of the formalism or tools used, the paper doesn't discuss any significant improvements to the existing approaches used to achieve (a) or (b).

>>> There are many different ways to combine symbolic and formal methods (as identified in the neuro-symbolic research field). Specifically, our work is located in the subfield of neuro-symbolic integration, which is the combination of ML methods with classical reasoning techniques.

The two novel contributions of our paper regarding neuro-symbolic integration that differentiate our work from the literature are: 1) integration of background-theory based deductive capability with symbolic regression and 2) introduction of (background-theory based) reasoning error measures from numerical data.

Moreover other minor contributions of the paper are: 1) use of the reasoning error measures to distinguish between background theories; 2) use of the reasoning error measures to perform dependence analysis; and 3) use of existential variables to represent numerical constants.

None of these techniques/ideas can be found in the literature (see following answer for more details).

As far as I know the idea was first proposed in the logic guided machine learning paper Scott et al. at AAAI 2020 and Ashok et al. at AAAI 2021, cited by the authors of this paper. However, the authors of this paper do not give enough credit to the Scott et al. or the Ashok et al. papers. In particular, their abstract needs to refer to both Scott et al. and Ashok et al., in order for the paper to be acceptable.

>>> **We substantially differentiate from the idea proposed by Scott et al. at AAAI 2020 and Ashok et al. at AAAI 2021. In their work they consider the integration of constraints based deduction, while we use background-theory based deduction. In simple words, in their work they check that a set of constraints is satisfied by the formula induced by the data. The usage of logic constraints to help with the discovery of formulas from numerical data has been considered in Kubalik et al. (2020) [3], Kubalik et al. (2021)[4], Engle and Sahinidis (2021) [5] in addition to Ashok et al. 2020 [1] and Scott et al. 2021 [2]. More generally, the usage of logic constraints as part of ML tools saw an increase in popularity from 2018: e.g. using the logic constraints violation as part of the loss of a NN (e.g. Xu et al. (2018) [6], Wang and Pan (2020) [7]) or with other techniques (e.g. Li and Srikumar (2019) [8], Daniele and Serafini (2020) [9], Xie et al (2019) [10], Li et al. (2019) [11]), among many others.**

Given a formula extracted from the data f' and a set of constraints C , in Scott et al. [2] and Ashok et al. [1] the authors want to prove that $f' \rightarrow C$, that is they wish to check that the formula satisfies certain constraints. The logic constraints they consider are constraints describing the functional form f' : “auxiliary truth = theorems or invariants over f' ” (first page of Scott et al. [2]). This is equivalent to what we call “constraints K ” at page 6/8 of the Methods part of our paper. These constraints are not part of our background theory and are instead included in the Modeler Preferences M (see page 3/8 of the main paper) or are a part of the definition of the hypothesis class C (page 2/3 of the main paper). We do not claim novelty in the usage of the constraints K . Constraint satisfaction can be helpful as a pre-processing step, and has been used in several works (see above) for improving the quality of hypotheses generated by SR.

Our approach is quite different in the sense that we want to prove that $C \rightarrow f'$ (see equation 8 and equation 9 in the Method part), that is the derivation of the formula from first principles. More precisely, we prove that $B \rightarrow f'$, where B is the set of background theory axioms. The axioms in B are fundamentally different from the axioms in K :

- 1) The axioms in K involve only variables that are present either in the input or output of the function f that we want to learn.

The axioms in B might contain some of the variables present in the input or output of f , but they can also: a) contain additional auxiliary variables not seen in the data and do not appear in the final functional form f ; b) contain only such auxiliary variables and none of the variables present in the data (input or output of f).

These axioms cannot be used by the approach by Scott et al. and Ashok et al.

An example from the Langmuir problem: the function to learn is of the form $f(p)=q$; The input/and output variable are the set $\{q,p\}$; An axiom in K is “ $\forall (p > 0)$ ”

$(f(p) > 0)$ ”; an axiom in B is “ $S_0 = S + Sa$ ” [The full list of the constraints in K for Langmuir’s problem can be found in section “*Thermodynamic constraints*” at page 2/20 of the the supplementary material. The full list of axioms in B for the same problem can be found at page 5/8 of the Method (axioms L1 to L5)].

- 2) The axioms in K describe the behavior of the function f , while the axioms in B do not describe the behavior of the function f , while they describe the behavior of other laws involved in the phenomena.

For example, given a function $f(x) = y$, an axiom in K might describe limiting behavior of f (e.g., when $x \rightarrow \infty$), while an axiom in B could be the conservation of energy law, that must be respected, but it does not involve x or y as variables.

Moreover, Scott et al. and Ashok et al. use a notion of distance, that is the degree of satisfaction of the constraints in K. This is similar to the works [3,4,5] etc.

Our reasoning distance is fundamentally different: it measures the distance from the function f' induced by the data and the correct function f , that we are trying to learn, that is implicitly defined by the background theory axioms in B.

To conclude, we are the first in the literature, to the best of our knowledge, to define and introduce the following:

- 1) integration of background-theory based (not constraints based) deductive capability with a ML method (a specific SR method, but this would work with any other equivalent method)
- 2) introduction of (background-theory based) reasoning error measures from numerical data
- 3) using the reasoning error measures to distinguish between background theories
- 4) use of the reasoning error measures to perform dependence analysis
- 5) use of existential variables to represent numerical constants in formulas obtained from symbolic regression

We already cited Ashok et al. 2020 and Scott et al. 2021 in the introduction, as representative of all these methods, and also refer to the fact that prior work has addressed constraint satisfaction in the abstract.

We added citations to other related works (in addition to the above two papers) in the Methods part, where we talk about the constraints K (page 6/8, end of Langmuir’s adsorption equation section).

Moreover we expanded the section “Additional related work” in the Supplementary material (page 4/20) with a discussion of these methods and the difference with our work in more details.

References:

- [1] Ashok et al. 2020 “LGML: Logic Guided Machine Learning (Student Abstract)”
- [2] Scott et al. 2021 “Logic Guided Genetic Algorithms”

- [3] Kubalik, et al. (2020) “Symbolic regression driven by training data and prior knowledge”
- [4] Kubalik, et al. (2021) “Multi-objective symbolic regression for physics-aware dynamic modeling”
- [5] Engle and Sahinidis (2021) “Deterministic symbolic regression with derivative information: General methodology and application to equations of state.”
- [6] Xu et al. (2018) “A Semantic Loss Function for Deep Learning with Symbolic Knowledge”
- [7] Wang and Pan (2020) “Integrating Deep Learning with Logic Fusion for Information Extraction”
- [8] Li and Srikumar (2019) “Augmenting Neural Networks with First-order Logic”
- [9] Daniele and Serafini (2020) “Neural Networks Enhancement through Prior Logical Knowledge”
- [10] Xie et al (2019) “Embedding Symbolic Knowledge into Deep Networks”
- [11] Li et al. (2019) “A Logic-Driven Framework for Consistency of Neural Models”

The main novelty of the paper, in my opinion, is the specific approach described by the authors to combine symbolic regression and automated theorem provers (specifically, MILP) to evaluate the quality of expressions generated the the symbolic regression engine. Thus, the paper can perhaps be best thought of as a "toy-model" describing a powerful idea already explored in previous research.

>>> **See answer above**

Q1: How is your work conceptually different from the above-mentioned papers?

>>> **See answer above**

Q2: Can you go beyond toy problems?

>>> **The problems we tackle are not toy problems in our opinion. Firstly, they all involve real data. In many common datasets in the literature, the data is synthetic (computer generated) with (or without) the addition of noise. When noise is synthetic, it may not match the noise behavior in real data. Such synthetic datasets are used in AI-Feynman, for example. For this reason, working with real data is rather different and nontrivial, requiring a more complex design. Secondly, even though the problems we study have few variables (in the symbolic regression problem), existing learning methods are unable to distinguish between multiple expressions that fit the data and have similar levels of complexity and numerical error.**

Quality of Writing

The paper is well-written with excellent quality graphical representations describing the overall idea. However, the paper falls short when it comes providing sufficient quantitative evidence (both in terms of the scale of the experiments and the choice of systems

used for experimentation) to allow to reader to discern the utility of the proposed approach on realistic examples.

>>>

[scale of the experiments]

We believe the examples we provide are still “realistic” in the sense that they are three real-world problems, involving only real (noisy) data, and existing methods are not capable of distinguishing between candidate formulas that have similar levels of numerical error and complexity.

Our goal is to show the novel capability of our system (to distinguish between candidate formulas) on this type of noisy data. Thus a quantitative comparison, of our system as a whole, on big simulated datasets or against methods that work only on synthetic data is not essential to demonstrate the novelty of our work. Moreover, another feature of our system that we wish to show is that it does not require a huge amount of data, but can work with only a few data points, which is often relevant in scientific discovery. Nonetheless, we have added a section explaining the scalability of our symbolic regression method in the subsection “Scaling” in the supplementary material.

[choice of systems]

The tools were chosen based on the requirements: e.g. Keymaera is the only reasoner that combines first order logic with a fast computation of arithmetics (based on Mathematica) and differential equations. We choose an MINLP based approach and the BARON MINLP solver in order to be able to obtain provably optimal symbolic regression solutions for each level of expression complexity (assuming a sufficient amount of running time).

To conclude, it is not uncommon in the literature to focus only on few problems, especially when they involve real data: some examples are “Advancing mathematics by guiding human intuition with AI”, A.Davies et al. (they focus on two mathematical problems) and “AI-Copernicus” R. Iten et al. (they focus on four physics problems).

Concerns

The numerical errors reported in the paper are quite large (~1e-3).

>>>

As the data is real numerical data, and not simulated data, there are measurement errors, and we do not believe it is possible to get lower numerical errors while fitting functions, unless we overfit. In fact, for the Einstein dataset, the error w.r.t. Einstein’s time dilation formula (the true formula) the L2 and L’infty absolute error are essentially the same as the approximate formula $-0.00563v^2$. For example, the L’infty absolute error w.r.t. the true formula is 0.319.

Further, it seems like the more complex the expressions

(cf. solar and binary stars in Table 1) lower are the average case numerical and reasoning errors. Is this approach, then, likely to 'over-fit', despite the use of domain knowledge?

>>> A standard approach is to plot the expression complexity and numerical error, and choose solutions from the Pareto front (which is what we do to obtain the candidate solutions in the paper). Our symbolic expression module (see description in the supplementary methods part of the paper) will obtain multiple expressions with differing levels of complexity. We then use the reasoning module to try to distinguish between these multiple expressions.

In particular, how do you decide between two candidate expressions, one with lower reasoning error and higher numerical error vs an expression with higher reasoning error and lower numerical error?

>>> In general the reasoning error has priority in our work, especially when we are confident that the background theory is complete and consistent. We believe a formula that has a higher numerical error and does not violate fundamental laws of nature is better than a formula that represents perfectly the available data but is not derivable. The data we are using are taken from real measurements with error; formulas with very low error in the data are likely modeling the noise as well. The formula with low reasoning error instead will generalize better on unseen data points. Another metric we considered is the complexity of the generated formula (lower complexity is preferred, for the same level of numerical error).

Further, for deriving Kepler's law, the paper uses Newton's law. What happens you remove that axiom? How do your errors change?

>>> In this work we assume that the background theory is consistent and complete. If we remove Newton's law, we will obtain an incomplete background theory (since Newton's law is a necessary axiom to prove Kepler's law).

In an incomplete theory (that is, without all necessary axioms) the problem is underspecified and thus we cannot guarantee that the derivable formula (for the variable of interest, that in Kepler case is the period P) is unique. This allows an infinite number of expressions for P to be consistent with the remaining axioms.

In this scenario the reasoning error is likely to be very large, or even unbounded. This is because the reasoning error reflects the error of the derivable formula with the largest error. For this reason it is important to have a complete background theory. In the case this is not possible we can use other reasoning techniques to try to complete it, such as abductive reasoning, which is part of our envisioned system (see Fig 3 of the main part of the paper and more details in the new section "System limitations and future improvements" in the Method part of the paper).

More generally, what are the minimum number of axioms required to distinguish between the candidate expressions in the experiments reported? This can be a very powerful additional line of research that

I think the authors ought to consider.

>>> **In our work we assume that the background theory is complete (it contains all the axioms necessary to comprehensively explain the phenomena under consideration) and consistent (the axioms do not contradict one another). This means that the background theory contains at least the minimal set of axioms that allows one to prove the correct formula. This minimal set is different and varies in cardinality for each given problem. The background theory can contain additional axioms that are not strictly necessary to prove the formula as long as they define a consistent theory. An interesting line of future research is to use abduction to fill the gaps when the background theory is not complete. We do not have this already integrated with our system, but this is part of our vision (see Fig. 3 of the main paper).**

Potential long-term impact

The idea proposed in the paper could be of significant long-term potential impact; however, as it stands, it is likely not mature enough to contribute to scientific progress. The question then, is whether such progress is possible.

Symbolic regression is still in its early stages of development and it might not be unreasonable to assume they will improve with time and be able to derive increasingly complex expressions with higher numerical accuracy. However, can automated theorem provers however, keep up with the associated complexity?

>>> **The complexity of the reasoning task is higher compared to symbolic regression: deductive inference in first order logic with arithmetic is not even decidable. However, there are many recent (in the last few years) developments in the neuro-symbolic area, that use deep-learning techniques to enhance standard theorem proving (e.g. we mentioned in the introduction the use of reinforcement learning to guide the search of a theorem prover “A deep reinforcement learning based approach to learning transferable proof guidance strategies” by Crouse, M. et al.). We envision that the performance and capability (in terms of speed and expressivity) of theorem provers will improve with time, just as symbolic regression solvers have improved a lot over time. We have added a section “System limitations and future improvements” discussing these issues in the Methods part of the paper.**

Most physical phenomena are often described using expressions that are simple closed form algebraic expressions. For example, underlying laws are often described in the language of probabilistic models or partial differential equations. If the idea is to use such a method for scientific discovery, in my opinion, it will necessary for both symbolic regression to be able to derive such expressions and automated theorem provers to be able to reason about them.

>>> Indeed this is a limitation of the tools used in the current system. We added a paragraph discussing these issues in detail in the Methods part of the paper (see section “System limitations and future improvements”). Keymaera is able to handle differential equations (only in time). This is one of the reasons we chose this tool, and it shows the potential of theorem provers in this regard. We think that in the future such methods will be developed.

Overall comment

The paper proposes a novel idea to combining machine learning and formal reasoning, based on previous work. However, it shows insufficient experimental evidence to convince the reader of its utility for real scientific problems. I wouldn't be inclined to publish this in Nature in its current form. Here are concrete suggestions for improvement:

* Cite previous work appropriately and prominently, including in the abstract

>>> We added the appropriate citations and expanded the section “Additional related work” in the supplementary material (see answers above for more details)

* Discuss limits of the method in detail

>>> We added a new section with details of the limitations of our system (see the last section “System limitations and future improvements” of the Methods part of the paper)

* Provide detailed empirical comparisons with other symbolic regression approaches such as AI Feynman, LGML, LGGA, Wetzel et al., Eureqa, TuringBot etc.

>>> We have run some of the suggested systems (and a few additional ones) on our datasets with real data. Of the systems mentioned above:

- **we were able to obtain and run the code of TuringBot and AI-Feynman**
- **we considered LGGA as well, but this was only possible with Langmuir's problem, since the system requires constraints on the functional form of f . We defined such constraints only for Langmuir's problem (what we called constraints K). However, the only constraint that is supported by LGGA is $f(0)=0$. The other constraints we use (monotonicity and condition at the limit) are not currently supported by LGGA. This didn't lead to good results so for this reason we did not report them.**
- **For the remaining (mentioned above) systems, either a commercial license is required or the code is not available.**
- **We included a comparison with additional systems such as PySR and Bayesian symbolic regression.**

In most cases, none of the systems is able to identify the right formula. Although some of them produce the formula in the top10 candidates, they have no tool to identify the right one within this list. We enhanced the systems with some standard methods to select the best candidate, such as computing the Pareto front and identifying knee points on it. We added the details of these experiments at the end of the Supplementary Material in the new section “Comparison with other systems”.

* Try adding different axioms to see whether the system can learn a new expression with minimal domain knowledge (where is the boundary for specific cases?)

>>> **In this work we assume that the background theory is consistent and complete. This is the minimal set of domain knowledge. There might be different minimal background theory sets, see for example the discussion on the Relativity problem where we consider a background theory with Newtonian assumptions and another one with Relativistic assumptions. Both background theories induce a unique solution. However the two background theories are not consistent within each other, since one assumes the speed of light is constant and the second one does not. These two facts are in contradiction to each other. The two different background theories induce different reasoning errors on the functional forms extracted from the data.**

* The numerical errors in the work presented are high. Can they be Lowered?

>>> **Please see the response to the first concern, where we point out that there is nontrivial amount of measurement error in some of the datasets, and even the “true” formula has significant error.**

* More experiments need to be done to better understand scalability (in terms of the length and complexity of symbolic expressions learnt)

>>> **We have added a new section on scaling and convergence for our symbolic regression module in the section titled “Scaling” in the supplementary material. We earlier described in detail how we solve symbolic regression problems from data in the supplementary material. In particular, we explained how we create multiple generalized expression trees, and solve a mixed-integer nonlinear programming (MINLP) problem per “gentree”. In the new section on scaling, we note that even a single MINLP can take a lot of time. We also added Figures 3 and 4 to illustrate convergence in two instances.**

We think that the scalability of the reasoning module is much harder to quantify. It is not clear how the “size” of a theorem-proving problem should be defined. There are several parameters that may affect the size: for example, the number and length of the axioms, the number of distinct predicates, functions and constants, the number of quantifiers (or the number of alternations between existential and universal quantifiers), etc. Moreover, there are several ways to write the same set of axioms: connecting all of them as a single rule (that is the conjunction of all the axioms), or by introducing new auxiliary predicates and variables to split the axioms and thus reducing their length.

Reviewer #3

(Remarks to the Author):

Needless to say, the topic covered in the paper is of enormous scientific interest! The paper creates the conditions for stimulating the reading and I also think that it comes at the right time in the overall research on Machine Learning. The examples that are chosen for claiming the generality of the proposed inferential mechanism are quite different, which leads [me] to believe that we are in front of a paper which can offer solid foundations in this field.

Unfortunately, the curious reader is likely to be quite disappointed when trying to disclose and understand the basic ideas.

Fig. 3 is nice, yet it is not supported by specific technical descriptions in the "Methods" section. >>> **Fig.3 represents the envisioned system. In this paper we describe in detail only the integration of the components represented by solid lines in the Figure. The components represented with dotted lines have not been integrated/developed yet. We added a more detailed discussion of the whole system, its limitations and current and future components in the section "System limitations and future improvements" at the end of the Methods part of the paper.**

The inferential mechanisms that emerge from data and, especially, from symbolic knowledge are only superficially described. One very much would like to appreciate [h]ow symbolic descriptions of the base knowledge that activates the inferential process and, particularly, how data promotes different inferential paths.

>>> **In the Methods part of the paper and in the supplementary material we give a detailed description of our symbolic regression module which creates symbolic expressions from data, and how our reasoning module tries to check for the consistency of the derived symbolic expressions. Throughout our paper we refer to numerical measurements as "data" and distinguish these from "background knowledge". If one considers the latter as data, then changing it would lead to different inferential paths in the reasoning module during the consistency checking/derivation phase. However, though we control the symbolic expression generator (as we developed the code ourselves), we use an off-the-shelf reasoning engine, KeYmaera X, and we cannot control the internal search process, except through limited parameter settings.**

In the current form, the paper is far from providing a convincing general scheme that can be used to support inference in different research domain. Along with more technical details, the code availability could help to understand the actual value of the paper.

>>> We thank the reviewer for bringing up the code availability question and its outreach significance. The full code is now available at <https://github.com/IBM/AI-Descartes> repository. Outside the Main section (6 pages) that provides a high-level overview of the results and novel techniques, technical details of the proposed approach and experiments can be found in the Methods file (8 pages) as well as in the Supplementary Material file(20 pages), containing further elaboration of the proposed algorithms, and pseudo code). We hope that the reviewer will find the technical details given across the 3 parts of the paper to be sufficient.

REVIEWER COMMENTS

Reviewer #1 (Remarks to the Author):

The author properly addressed the issues raised. The effort on AI for Scientific Discovery in general is still in an infancy, the work presented here represents meaningful progress that should be useful for the community.

Reviewer #2 (Remarks to the Author):

Summary of the paper:

The authors aim to utilize a combination of symbolic regression and background knowledge of a problem domain to select candidate formulae that are consistent with the background knowledge and that have a sufficiently low error with respect to a derivable formula given only axioms known about the system, and low error with respect to data from an experiment that models the system in question. Specifically, the authors utilize a 4-tuple consisting of $\{B, C, D, M\}$. B models the domain knowledge as logic formulae that provide axioms concerning the input vector, x , and the desired symbolic model, y . They assume this background theory is complete (that is, that contains all necessary axioms to describe the system in question), and that it is consistent (that is, no logical contradictions arise from the axioms). C is called the "hypothesis class", a set of symbolic models defined by some grammar and with constraints to remove redundant expressions (they provide an example of such a constraint as the commutativity of addition). D is the data and is simply the set of data points that provide examples of the system. M is the set of modeller preferences that provide additional parameters on the model, such as bounds for accuracy.

Their approach consists of two main modules: a symbolic regression module, and a reasoning module. The symbolic regression module provides a set of candidate models that fit the data, and which are tested to determine if they satisfy the constraints provided by C , satisfy other parameters provided by M , and are tested for derivability from B . From here, the reasoning module may return one of the models if it is derivable from B . If none of the models are found to be derivable, a set of metrics are utilized to prune the set of candidate functions. Two error functions, epsilon and beta, are utilized here. Beta measures the distance between a function derived from the numerical data, f , and a second function derived solely from B , f_B . epsilon is then defined as the difference between f and the dataset. Noteworthy here is that f_B can be used to directly prove whether some f are not derivable from B by returning points satisfying B that do not fit in the model f .

Their symbolic regression solver is based on mixed-integer nonlinear programming (MINLP) utilizing (+, -, *, /, sqrt(), log(), exp()) as the operators to be used in the regression, as well as upper bounds on the complexity of the expression and number of constants not equal to 1. Their symbolic regression solver utilizes MINLP instances to find expressions that minimize error given a dataset. Notably, their solver allows for linear and nonlinear constraints, as well as dimensional consistency whenever physical dimensions of variables in the input vector are known.

To test their proposed model, they utilized data to derive expressions explaining Kepler's third law of planetary motion, Einstein's time-dilation formula, and Langmuir's absorption equation. They were able to successfully re-discover Kepler's third law, though they conceded that the differing magnitudes of masses on the planetary scale can make the problem more difficult. They were not able to successfully re-discover Einstein's equation for time-dilation, though they were able to select the formula that generalizes best, as well as identify a theory that better fits the data through the use of the reasoning module and alternative axioms (following "Newtonian" assumptions as opposed to relativistic ones). In the case of Langmuir's absorption equation, they related the material-dependent coefficients in the known equation with those present in the models generated by symbolic regression to logically prove their extracted formula.

Their model contains all aspects necessary for symbolic regression (data sets, grammar, operators, etc.) and for their logical reasoning (axioms, derivability, etc.), though it presently lacks the capability of abductive reasoning to suggest new axioms or to provide feedback on the experimental design to add additional constraints or conditions on the data. It also lacks the capability to utilize outside models to refine the generation of candidate hypotheses/functions. Though, the system proves itself quite capable given a very small dataset, as the Kepler dataset only consisted of 28 total data points in 4 dimensions (m_1 , m_2 , d , and p . These represent the masses of the two bodies, the distance between them, and the orbital period, respectively).

Novelty and comparison to previous work:

Over the past several decades, much work has been done in the field of symbolic regression and automated theorem provers separately. One of the very first work that combines the two (i.e., machine learning and domain knowledge for scientific knowledge discovery) is by Scott et al. (Logic Guided Machine Learning) and Ashok et al. (Logic Guided Genetic Algorithms). The authors cite such symbolic regression solvers as AIFeynman and others, including those that utilize genetic programming or those that utilize MINLP. In their symbolic regression module, the authors utilize a novel MINLP-based solver to generate candidate symbolic expressions for a given dataset. Similar approaches have been explored by many other authors in the years, but was initially developed as a globally optimal approach to generate solutions to symbolic regression problem instances by Austel et al. in 2017.

The authors also call attention to Automated Theorem Provers (ATPs), as they might be used to instead prove conjecture from a base theory. Though, approaches that purely utilize ATPs are quite computationally intensive, and the authors note that some instances are undecidable. They cite a paper by Grigoryev, Hirsch, & Pasechnik as an example where deriving models for inequalities over arithmetic and calculus operators prove particularly difficult. Though, some other authors have utilized machine-learning techniques to improve the performance of ATPs, such as those done by Crouse et al. and Fawzi et al. In this sense, the authors aim to replicate similar results by utilizing logic guidance in conjunction with regression methods to improve performance of symbolic regression.

The nearest approximation for prior works in the literature that aim to combine symbolic regression and logical models are the two works by Scott, Panju, and Ganesh (LGML) as well as the paper by Ashok et al. (LGGA). These two papers follow a similar approach in that they aim to combine the use of logical reasoning to aid in the process of symbolic regression. In the first paper, the authors present a novel approach to utilize a logic solver to refine the learning process in symbolic regression by augmenting the input dataset to be learned. This is achieved by providing counterexamples that satisfy an outside known property of the system, an "auxiliary truth", whenever the learned function is inconsistent with this truth. The second paper expands on this idea, allowing for genetic algorithms to utilize these auxiliary truths to improve data efficiency and accuracy of models derived with this method.

The methods described above differ from the authors of AIDescartes, however, as the authors of the LGGA and LGML papers utilize feedback loops that augment data to guide their learning module towards a model consistent with given auxiliary truths. The AIDescartes authors instead do not modify the dataset that is learned from, and utilize metrics from their reasoning module to determine derivability from known truths of the system. The authors of AIDescartes also note that these methods only consider constraints on the form of the function to be learned, while AIDescartes considers other truths and constraints that describe laws or unmeasured variables within the system, making the approach a novel one in the literature.

Additionally, the authors discuss other works succinctly in the main paper and at greater length in the supplemental material. The authors also provide credit to the works utilized directly in their method that were not created by them (e.g.: the use of KeyMaera X in their reasoning module). It might be worthwhile to include additional references to similar works in the sphere of application of logic to symbolic regression to draw contrasts between this method and other works at the time of publication, but the present work has sufficient mention and acknowledgement of previous work

However, the authors fail to cite any previous work on NeuroSymbolic AI, a field that aims to combine logical reasoning (e.g., via solvers) with machine learning. Please refer to the book by Garcez and Lamb to get a better idea of this field and cite appropriate work.

Experimental Results:

The results presented by the authors seem sufficient and trustworthy for their claims, though additional data would prove useful in further demonstrating accuracy and efficiency of their method. The experiments utilized to demonstrate the end-to-end capabilities of AIDescartes accomplish just that, though the provided comparison to select other methods of symbolic regression provide little more than verification of the experimental results. Additionally, the experiments selected seem very narrow in scope. Though these may have been chosen for the sake of algorithmic/run-time complexity, additional data for the performance of the method or supplemental examples/experiments would aid in demonstrating the breadth of the method (i.e.: "harder" instances of symbolic regression do not grind this method to a halt).

The design of the experiments do follow the scientific method, and allow for reasonable comparisons to be drawn between other methods in the literature and conclusions to be made regarding AIDescartes and its performance. The data supports that the method is in fact accurate and of the same quality as other works in the literature. The authors do honestly admit the limitations of their own work, as the method was unable to recover Einstein's time dilation formula in the second experiment. The authors also include the necessary data points used in the experiments, the parameters for other methods used for comparison, and the results of these other methods within the supplementary material. This transparency enables anyone to verify the experimental results, making the results arguably quite trustworthy.

Algorithmic Efficacy:

The presented method is certainly effective with respect to the size of the initial dataset presented, as the reasoning module is able to select the correct formula out of a set of candidates generated when a very small dataset is given to the algorithm. The authors express that the efficiency of the MINLP solver used during regression has complexity that grows exponentially as the expression to be searched for becomes larger or more complex (as a result of adding more allowed operators to the language, as the depth of trees generated from operators grows, etc.). For this reason, parallelism is exploited wherever possible to enable the solver to perform more efficiently. The two modules are also able to demonstrate end-to-end discovery of mathematical formulae utilizing few data points and domain knowledge of the problem to be modeled, which supports the original claim made by the authors in the introduction.

Potential impact:

This work may have a profound influence on future works that aim to study how logical reasoning can be applied to symbolic regression, as it seems that very few papers exist on this subject currently. This provides direct evidence that logical reasoning can be applied to symbolic regression to refine models derived, as well as to prove derivability (or lack thereof) of candidate models during the regression step. The authors also note that their work may be built upon to incorporate other features not present with the current method, such as the capability for abductive reasoning or knowledge of outside models. This work is thus likely to influence further research into the application of logic into symbolic regression, as well as into other novel methods to refine models generated.

Verdict: Major revision and resubmit

* As written, the paper demonstrates a good contribution to the greater scientific community with a slightly novel approach to symbolic regression that incorporates logical reasoning. However, they make very strong claims in their abstract and elsewhere that their work is a first step in combining machine learning (symbolic regression) and domain knowledge towards the goal of automating the scientific method. This is demonstrably false. Please see the LGML and LGGA that the authors do cite later in their paper. Please cite these works in the abstract and elsewhere and properly differentiate with your work. Please remove strong claims like "a crucial first step..." in the abstract. Replace it with a "building on previous work, our approach is a step in the direction of automating the scientific method."

* Please look up previous work on NeuroSymbolic AI and differentiate your work from this decades-old field.

* The experimental aspect, while a good step in the direction, is still quite weak. For example, in their work on LGML and LGGA, Scott et al. and Ashok et al. report being able to learn several equations from the Feynman book. By contrast, AI Descartes is only able to learn a few. It is not clear why this is the case.

Reviewer #4 (Remarks to the Author):

What are the noteworthy results?

The authors present the tool AI-Descartes -- a tool that leverages logic solvers and theorem provers with symbolic regression aimed for automated scientific discovery. The tool is quite intricate and leverages a feedback loop between these seemingly independent systems. The authors present an analysis on three problems, namely, Kepler's third law, time dilation, Langmuir's adsorption equation.

Will the work be of significance to the field and related fields? How does it compare to the established literature? If the work is not original, please provide relevant references.

The tool is extremely relevant in the field of automatic scientific discovery and shows how it can discover three known equations in physics. However, the tool seems quite focused, and it is unclear if it will have cross-disciplinary applications. The tool is closely related to two other works by Scott et al. and Ashok et al.

It is hard for me to interpret the rebuttal for Reviewer #2, as the response mislabels LGML and LGGA. Scott et al. was LGML and LGGA was Ashok et al. ([1],[2]). Even so, the response to the second question of reviewer #2 is a bit unclear. Particularly:

"in Scott et al. [2] and Ashok et al. [1] the authors want to prove that $f' \rightarrow C$,"..... Our approach is quite different in the sense that we want to prove that $C \rightarrow f'$ (see equation 8 and equation 9 in the Method part),

This does not make any sense to me, and this is not what either work achieves. There seems to be some misunderstanding here.

Does the work support the conclusions and claims, or is additional evidence needed?

All claims seem correct. However, it is not clear if its a state-of-the-art tool for symbolic discovery. Based on the empirical evaluations of tools like AI-Feynman and PySR, it seems these tools would also be able to solve the underlying symbolic regression problems considered in the paper. Is this not the case? What is the value-addition of AI-Decartes here?

The authors go into great depth on three examples and have made a case that the idea of the paper has some merit on these examples. However, just three benchmarks remain unjustified. The authors make a note of other tools such as AI-Feynman PySR but do not baseline against it. AI-Feynman provided a

significant dataset of several physics problems. There is a benchmark suite by William La Cava's group (<https://github.com/cavalab/srbench>). Furthermore, there is now a competition in this space at GECCO 2022.

Are there any flaws in the data analysis, interpretation and conclusions? Do these prohibit publication or require revision?

There are several modules in the figure architecture diagram that remain either unexplained or have an unclear impact.

What is the experimental design module? There is almost no commentary on this. However, it is an integral part of the tool. It seems similar to the Oracle Labeler in Scott et al. ? The oracle labeler is a major assumption made by that tool. This needs to be expanded on.

The abductive reasoning module sounds quite interesting, but it is unclear how it works from the paper. What "new axioms" did it propose in the three experiments? Did it find anything useful?

Is the methodology sound? Does the work meet the expected standards in your field?

The authors go into great depth on three examples. While I appreciate the author's argument in the rebuttal for only considering 3 benchmarks, it is still unclear to me if there is publication bias. Even just a few additional experiments (even if not in as such detail as before) would help.

I am unable to draw any conclusions on how AI-Decartes compares to state-of-the-art tools in this space.

There is also a benchmark suite provided by ca Furthermore, there have been recent competitions in this space,

Is there enough detail provided in the methods for the work to be reproduced?

A link to a github source code is provided in the rebuttal but not in the paper.

Why is the number of const part of the input? This is atypical for symbolic regression problems.

What is the experimental design module? There is almost no commentary on this. However, it is an integral part of the tool. It seems similar to the Oracle Labeler in Scott et al. ? The oracle labeler is a major assumption made by that tool. This needs to be expanded on.

Summary

Overall, I feel like the system behind AI-Descartes shows great promise. However, I agree with reviewers #2 and #3 that the empirical evaluation is unconvincing. It is not clear to me where the tool falls relative to state-of-the-art, and several components of the tool are not discussed, and the value-addition is unclear. The idea of the paper is quite similar to Scott et al. and Ashok et al., while there are some differences, the authors' explanation of the difference is not clear and believe to be incorrect.

TO THE REVIEWERS:

We sincerely thank the referees for carefully reading the manuscript.

The constructive suggestions have helped us to improve our paper further. We appreciate the reviewers finding that our work “represents meaningful progress that shall be useful for the community” (Reviewer 1), “may have a profound influence on future works that aim to study how logical reasoning can be applied to symbolic regression” (Reviewer 2) and “shows great promise” (Reviewer 3).

The major issues raised by the reviewers, and their resolution, are summarized as follows:

Q1: Does the fact that our system deals only with “small” problems render it useless in practice?

The underlying premise of this question is that problems involving simple formulas are well-solved.

We have performed extensive experimentation with 81 out of the 100 problems from the Feynman Symbolic Regression Database (FSRD) developed in the AI Feynman paper. Though we use the same independent and dependent variables, we modified the datasets to make them more similar to real data: we selected only 10 (from the beginning) data points (to reflect the fact that collecting data is expensive in real-life) and added normally distributed noise to the dependent variable value in the same way as is done in the AI Feynman paper (real data is noisy). The resulting dataset is still synthetic but is similar to the real-life datasets we focus on in our work in that there are few, noisy points. We then compare our Symbolic Regression module with state-of-the-art methods such as AI Feynman, PySR, Bayesian Machine Scientist, and TuringBot. The results are reported in section “Comparison with other systems” in the supplementary-material part of the paper (see Tables 13-15). We observe that, for 29/81 problems (~36%), none of the solvers are able to obtain the ground-truth formula.

Therefore, even for the seemingly simple and small (few term) formulas in the FSRD, SR is not well-solved when we consider datasets that mimic many real-life situations, with few data points and noise. Our Reasoning module can be very useful in such a situation by using prior knowledge to identify the correct formula (or its best approximation) within the candidates.

Thus, even without scaling to larger problems, we believe our system can be very useful when considering real data.

Q2: Does our symbolic regression module advance the state-of-the-art in symbolic regression?

When considering datasets resembling real data, AI Descartes is able to recover the correct formula in the largest number of cases, outperforming all the tested state-of-the-art SR methods

by a margin of at least 10% (see Tables 13-15 in the Supplementary Material). For this reason we conclude that AI Descartes does indeed advance the state of the art in the context of the physics problems in FSRD. We note that AI Feynman paper reports recovering all 100 formulas; however, the only experiments performed either use many data points, or a small number of noise-free data points. In our more realistic setting, mimicking real data, AI Feynman does not perform as well, and our proposed method outperforms it with an improvement of 20%.

We addressed in section “Comparison with other systems” the reviewers' concern (2 and 4) about the empirical evaluation providing a clear comparison with the state of the art. This section can be found in the supplementary-material part of the paper, starting at page 18.

Q3: Does our combined regression + reasoning module advance the state-of-the-art in scientific discovery?

We believe we have advanced the state of the art in SR (see our answer to Q2). We also believe we advanced the state of the art in combined reasoning and regression functionality. We are the first to propose the use of general logical reasoning (not only using constraints (e.g., monotonicity) on the functional form of the output formula extracted from the data to distinguish between candidate formulas. Thus, we allow integration of general knowledge about the environment (e.g. fundamental laws such as conservation of energy) that are not handled by current state-of-the-art methods. Many methods, including AI-Feynman, do not have a clear way to pick the right formula, even if it is listed among the possible candidates. Moreover, when a method is not able to generate the right formula, we are able to provide guidance for the formula that is the best approximation from among the “wrong” ones. We believe this innovation will have an interesting impact in the field of scientific discovery.

We also performed new experiments to demonstrate the performance of the reasoning module on problems from FSRD: we extracted the background theory axioms for five problems, and for each problem we chose only 10 points per dataset and added 1% noise. As shown in Table 15, for all five problems, we were able to derive the right formula, among the candidates produced by our SR module.

The FSR database does not provide an associated background theory; thus, we annotated only five problems, as annotation is very time consuming and requires expert knowledge (see answer to Q4).

Q4: Why do we not run experiments on more (available) datasets?

As we mentioned in Q2 and Q3, we have now tested our symbolic regression module on 81 new problems, and our complete discovery framework on five new problems. However, the data we used for these 81 + 5 instances is synthetic but noisy.

In order to test our model on more real-world examples, a prerequisite is the availability of a formalized background theory and a corresponding dataset with data collected by real-world

measurements. Unfortunately, this type of data currently is not easy to find for the following two reasons:

- Lack of real-world datasets: The datasets that are available for scientific discovery (e.g. physics, chemistry etc. domains) are mostly synthetic. Real data contains a larger amount of noise, and the noise is not as regular as in the synthetic data. Moreover, real data can be expensive to obtain (because data collection is costly and often difficult): thus there are fewer datasets with real measurements compared to synthetic ones where the number of points can be chosen by the user. Most of the state-of-the-art methods are tailored to solve synthetic datasets, often performing very poorly on real data; for example, adding noise and reducing the number of data points leads to a drop in the performance of AI-Feynman by 60%.
- Open problem of axiomatization of physics: Our work is based on the availability of problems that have an associated theory whose axioms and conjectures are machine-readable and thus amenable to automated solvers such as KeYmaera X (usually restricted to algebraic functions and first-order logical reasoning with equations). This requires time and expertise and is generally done by hand and is much more complex than dealing with synthetic datasets such as FSRD. In this work, we formalized all of these theories ourselves, by consulting with experts in the field. Since Hilbert's 6th problem (*David Hilbert, "Mathematical problems", Bull. Amer. Math. Soc. 8, 1902*), many have discussed the axiomatization of physics, which remains an open problem (*Alexander N. Gorban, "Hilbert's sixth problem: the endless road to rigour.", Philosophical Transactions of the Royal Society A: Mathematical, Physical and Engineering Sciences, 2018*). Our idea is in line with the recent interest in considering scientific theories as collections of proofs (Paleo 2012 – Physics and Proof Theory; Davis 2019 – Proof Verification Technology and Elementary Physics). Thus, we are trying to solve the more fundamental problem of theory-and-data-driven automated scientific discovery, which is radically different from the current machine learning approaches, which are usually benchmark driven. Finally, we believe that our tool will encourage people to formalize more scientific theories, creating new datasets, similarly to the six datasets we provide for the three problems we considered in our work. Moreover, given the fact that AI-Descartes is in its infancy, more work is possible in this direction, by e.g. leveraging richer theorem provers, or improving/substituting on the components, that could open our tool up to richer applications.

We answer in more detail the single reviewer questions in what follows. Our comments are preceded by >>> and appear in **blue bold font**. All the changes in the paper are in blue color.

Reviewer #1

The author properly addressed the issues raised. The effort on AI for Scientific Discovery in general is still in an infancy, the work presented here represents meaningful progress that shall be useful for the community.

>>> Thank you. We appreciate your effort in reading the paper and your positive comments.

Reviewer #2

Summary of the paper:

The authors aim to utilize a combination of symbolic regression and background knowledge of a problem domain to select candidate formulae that are consistent with the background knowledge and that have a sufficiently low error with respect to a derivable formula given only axioms known about the system, and low error with respect to data from an experiment that models the system in question. Specifically, the authors utilize a 4-tuple consisting of $\{B, C, D, M\}$. B models the domain knowledge as logic formulae that provide axioms concerning the input vector, x , and the desired symbolic model, y . They assume this background theory is complete (that is, that it contains all necessary axioms to describe the system in question), and that it is consistent (that is, no logical contradictions arise from the axioms). C is called the "hypothesis class", a set of symbolic models defined by some grammar and with constraints to remove redundant expressions (they provide an example of such a constraint as the commutativity of addition). D is the data and is simply the set of data points that provide examples of the system. M is the set of modeller preferences that provide additional parameters on the model, such as bounds for accuracy.

Their approach consists of two main modules: a symbolic regression module, and a reasoning module. The symbolic regression module provides a set of candidate models that fit the data, and which are tested to determine if they satisfy the constraints provided by C , satisfy other parameters provided by M , and are tested for derivability from B . From here, the reasoning module may return one of the models if it is derivable from B . If none of the models are found to be derivable, a set of metrics are utilized to prune the set of candidate functions. Two error functions, epsilon and beta, are utilized here. Beta measures the distance between a function derived from the numerical data, f , and a second function derived solely from B , f_B . epsilon is then defined as the difference between f and the dataset. Noteworthy here is that f_B can be used to directly prove whether some f are not derivable from B by returning points satisfying B that do not fit in the model f .

Their symbolic regression solver is based on mixed-integer nonlinear programming (MINLP) utilizing $(+, -, *, /, \text{sqrt}(), \text{log}(), \text{exp}())$ as the operators to be used in the regression, as well as upper bounds on the complexity of the expression and number of constants not equal to 1. Their symbolic regression solver utilizes MINLP instances to find expressions that minimize error given a dataset. Notably, their solver allows for linear and nonlinear constraints, as well as dimensional consistency whenever physical dimensions of variables in the input vector are known.

To test their proposed model, they utilized data to derive expressions explaining Kepler's third law of planetary motion, Einstein's time-dilation formula, and Langmuir's absorption equation. They were able to successfully re-discover Kepler's third law, though they conceded that the differing magnitudes of masses on the planetary scale can make the problem more difficult.

They were not able to successfully re-discover Einstein's equation for time-dilation, though they were able to select the formula that generalizes best, as well as identify a theory that better fits the data through the use of the reasoning module and alternative axioms (following "Newtonian" assumptions as opposed to relativistic ones). In the case of Langmuir's absorption equation, they related the material-dependent coefficients in the known equation with those present in the models generated by symbolic regression to logically prove their extracted formula.

Their model contains all aspects necessary for symbolic regression (data sets, grammar, operators, etc.) and for their logical reasoning (axioms, derivability, etc.), though it presently lacks the capability of abductive reasoning to suggest new axioms or to provide feedback on the experimental design to add additional constraints or conditions on the data. It also lacks the capability to utilize outside models to refine the generation of candidate hypotheses/functions. Though, the system proves itself quite capable given a very small dataset, as the Kepler dataset only consisted of 28 total data points in 4 dimensions (m_1 , m_2 , d , and p . These represent the masses of the two bodies, the distance between them, and the orbital period, respectively).

Novelty and comparison to previous work:

Over the past several decades, much work has been done in the field of symbolic regression and automated theorem provers separately. One of the very first work that combines the two (i.e., machine learning and domain knowledge for scientific knowledge discovery) is by Scott et al. (Logic Guided Machine Learning) and Ashok et al. (Logic Guided Genetic Algorithms). The authors cite such symbolic regression solvers as AIFeynman and others, including those that utilize genetic programming or those that utilize MINLP. In their symbolic regression module, the authors utilize a novel MINLP-based solver to generate candidate symbolic expressions for a given dataset. Similar approaches have been explored by many other authors in the years, but was initially developed as a globally optimal approach to generate solutions to symbolic regression problem instances by Austel et al. in 2017.

The authors also call attention to Automated Theorem Provers (ATPs), as they might be used to instead prove conjecture from a base theory. Though, approaches that purely utilize ATPs are quite computationally intensive, and the authors note that some instances are undecidable. They cite a paper by Grigoryev, Hirsch, & Pasechnik as an example where deriving models for inequalities over arithmetic and calculus operators prove particularly difficult. Though, some other authors have utilized machine-learning techniques to improve the performance of ATPs, such as those done by Crouse et al. and Fawzi et al. In this sense, the authors aim to replicate similar results by utilizing logic guidance in conjunction with regression methods to improve performance of symbolic regression.

The nearest prior art that aims to combine symbolic regression and logical models are the two works by Scott, Panju, and Ganesh (LGML) as well as the paper by Ashok et al. (LGGA). These two papers follow a similar approach in that they aim to combine the use of logical reasoning to aid in the process of symbolic regression. In the first paper, the authors present a novel approach to utilize a logic solver to refine the learning process in symbolic regression by augmenting the input dataset to be learned. This is achieved by providing counterexamples that

satisfy an outside known property of the system, an "auxiliary truth", whenever the learned function is inconsistent with this truth. The second paper expands on this idea, allowing for genetic algorithms to utilize these auxiliary truths to improve data efficiency and accuracy of models derived with this method.

The methods described above differ from the authors of AIDescartes, however, as the authors of the LGGA and LGML papers utilize feedback loops that augment data to guide their learning module towards a model consistent with given auxiliary truths. The AIDescartes authors instead do not modify the dataset that is learned from, and utilize metrics from their reasoning module to determine derivability from known truths of the system. The authors of AIDescartes also note that these methods only consider constraints on the form of the function to be learned, while AIDescartes considers other truths and constraints that describe laws or unmeasured variables within the system, making the approach a novel one in the literature.

Additionally, the authors discuss other works succinctly in the main paper and at greater length in the supplemental material. The authors also provide credit to the works utilized directly in their method that were not created by them (e.g.: the use of KeyMaera X in their reasoning module). It might be worthwhile to include additional references to similar works in the sphere of application of logic to symbolic regression to draw contrasts between this method and other works at the time of publication, but the present work has sufficient mention and acknowledgement of previous work.

However, the authors fail to cite any previous work on NeuroSymbolic AI, a field that aims to combine logical reasoning (e.g., via solvers) with machine learning. Please refer to the book by Garcez and Lamb to get a better idea of this field and cite appropriate work.

>>> We added more references for relevant Neuro Symbolic AI works. In particular we added references to the subfields that we thought are pertinent to our work: incorporation of constraints in NN, program synthesis, neuro-symbolic inductive logic programming and rule induction. If there are any other specific subfields of Neuro-Symbolic AI that you find relevant, please let us know.

Experimental Results:

The results presented by the authors seem sufficient and trustworthy for their claims, though additional data would prove useful in further demonstrating accuracy and efficiency of their method. The experiments utilized to demonstrate the end-to-end capabilities of AIDescartes accomplish just that, though the provided comparison to select other methods of symbolic regression provide little more than verification of the experimental results. Additionally, the experiments selected seem very narrow in scope. Though these may have been chosen for the sake of algorithmic/run-time complexity, additional data for the performance of the method or supplemental examples/experiments would aid in demonstrating the breadth of the method (i.e.: "harder" instances of symbolic regression do not grind this method to a halt).

>>> (See answers to Q1 and Q2 in the general response.)

We performed more experiments to compare with the state-of-the-art methods for scientific discovery. Other than comparing our system with some state-of-the-art methods on the three real-life problems that we considered in our work, we also

performed a comparison on 81 problems out of 100 from the Feynman Symbolic Regression Database (FSRD). We excluded 19 formulas that contain trigonometric functions, which our SR solver cannot handle (see section on the limitation of our tool). For each of the 81 problems, we chose 10 data points from each (synthetic) dataset and added 1% error (as described in the AI-Feynman work), creating small, noisy datasets that resemble real-life data (which is the focus of our paper). We show the performance of the state-of-the-art methods in Table 13 and Table 14.

The design of the experiments do follow the scientific method, and allow for reasonable comparisons to be drawn between other methods in the literature and conclusions to be made regarding AIDescartes and its performance. The data supports that the method is in fact accurate and of the same quality as other works in the literature. The authors do honestly admit the limitations of their own work, as the method was unable to recover Einstein's time dilation formula in the second experiment. The authors also include the necessary data points used in the experiments, the parameters for other methods used for comparison, and the results of these other methods within the supplementary material. This transparency enables anyone to verify the experimental results, making the results arguably quite trustworthy.

Algorithmic Efficacy:

The presented method is certainly effective with respect to the size of the initial dataset presented, as the reasoning module is able to select the correct formula out of a set of candidates generated when a very small dataset is given to the algorithm. The authors express that the efficiency of the MINLP solver used during regression has complexity that grows exponentially as the expression to be searched for becomes larger or more complex (as a result of adding more allowed operators to the language, as the depth of trees generated from operators grows, etc.). For this reason, parallelism is exploited wherever possible to enable the solver to perform more efficiently. The two modules are also able to demonstrate end-to-end discovery of mathematical formulae utilizing few data points and domain knowledge of the problem to be modeled, which supports the original claim made by the authors in the introduction.

Potential impact:

This work may have a profound influence on future works that aim to study how logical reasoning can be applied to symbolic regression, as it seems that very few papers exist on this subject currently. This provides direct evidence that logical reasoning can be applied to symbolic regression to refine models derived, as well as to prove derivability (or lack thereof) of candidate models during the regression step. The authors also note that their work may be built upon to incorporate other features not present with the current method, such as the capability for abductive reasoning or knowledge of outside models. This work is thus likely to influence further research into the application of logic into symbolic regression, as well as into other novel methods to refine models generated.

Verdict: Major revision and resubmit

* As written, the paper demonstrates a good contribution to the greater scientific community with a slightly novel approach to symbolic regression that incorporates logical reasoning. However, they make very strong claims in their abstract and elsewhere that their work is a first step in combining machine learning (symbolic regression) and domain knowledge towards the goal of automating the scientific method. This is demonstrably false. Please see the LGML and LGGA that the authors do cite later in their paper. Please cite these works in the abstract and elsewhere and properly differentiate with your work. Please remove strong claims like "a crucial first step..." in the abstract. Replace it with a "building on previous work, our approach is a step in the direction of automating the scientific method."

>>> We reformulated the abstract following your advice, adding the appropriate citations. In the new version of the paper we are citing the two works you mention (LGML, LGGA) both in the Abstract, Main paper and Supplementary material. A detailed description of the differences between our work and LGML and LGGA can be found in the section "Additional related work" in the supplementary material. We also removed the strong claim "a crucial first step", from the abstract.

* Please look up previous work on NeuroSymbolic AI and differentiate your work from this decades-old field.

>>> We have added more references for relevant Neuro Symbolic AI works. In particular we added references to the subfield that we thought are pertinent in our context: incorporation of constraints in NN, program synthesis, neuro-symbolic inductive logic programming and rule induction. If there are other specific sub-fields of Neuro-Symbolic AI that you find relevant, please let us know.

* The experimental aspect, while a good step in the direction, is still quite weak. For example, in their work on LGML and LGGA, Scott et al. and Ashok et al. report being able to learn several equations from the Feynman book. By contrast, AI Descartes is only able to learn a few. It is not clear why this is the case.

>>> Please see our answer to questions Q1 and Q2 in the general response, where we explain that we performed an extensive set of new SR experiments comparing our code on problems from the Feynman Symbolic Regression Database (FSRD) with other well-known symbolic regression solvers, namely AI Feynman, PySR, Bayesian Machine Scientist, and TuringBot. We took the synthetic, zero error data from FSRD, chose only 10 data points, and then added 1% error to the dependent variable value. The purpose of this modification is to make the data resemble the real-life data we work on.

These tests conclusively demonstrate that existing pure symbolic regression techniques are simply not capable of obtaining the "correct" formula from data with high probability. We do not think this is because of the quality of the SR techniques we use, but because of the fact that there are a very large number of functions with very different functional forms that have a similar error level on a noisy dataset with few points, and SR cannot distinguish between these functions.

Reviewer #4

What are the noteworthy results?

The authors present the tool AI-Descartes -- a tool that leverages logic solvers and theorem provers with symbolic regression aimed for automated scientific discovery. The tool is quite intricate and leverages a feedback loop between these seemingly independent systems. The authors present an analysis on three problems, namely, Kepler's third law, time dilation, Langmuir's adsorption equation.

Will the work be of significance to the field and related fields? How does it compare to the established literature? If the work is not original, please provide relevant references.

The tool is extremely relevant in the field of automatic scientific discovery and shows how it can discover three known equations in physics.

However, the tool seems quite focused, and it is unclear if it will have cross-disciplinary applications. The tool is closely related to two other works by Scott et al. and Ashok et al. It is hard for me to interpret the rebuttal for Reviewer #2, as the response mislabels LGML and LGGA. Scott et al. was LGML and LGGA was Ashok et al. ([1],[2]). Even so, the response to the second question of reviewer #2 is a bit unclear. Particularly:

"in Scott et al. [2] and Ashok et al. [1] the authors want to prove that $f \rightarrow C$,"..... Our approach is quite different in the sense that we want to prove that $C \rightarrow f$ (see equation 8 and equation 9 in the Method part),

This does not make any sense to me, and this is not what either work achieves. There seems to be some misunderstanding here.

>>> LGML and LGGA use what they call Auxiliary Truths (AT). Please note that LGGA is an extension of LGML. In both these works, auxiliary truth means "mathematical expressions that capture domain specific knowledge or simple properties of an unknown function" (see page 3 of LGGA) which is a "logical equation in terms of the feature space and the unknown function f " (see page 2 of LGML). This means that the auxiliary truths they consider are equations or inequalities on the variables present in the data.

Our work is very different in the sense that we do not consider these types of constraints for our reasoning module, while we consider general laws that describe the environment, often not including any of the variables present in the data. The axioms we consider do not impose constraints on the functional form of the function we want to learn. More importantly, since fundamentally different, our axioms cannot be used by methods such as LGML or LGGA.

In both LGML and LGGA the authors deduce formulas from the data via Symbolic Regression algorithm/Neural Network, or via a genetic algorithm, respectively; they then check if the functional form of the formula satisfies a set of properties, namely, the mathematical constraints termed Auxiliary Truths. Given an AT property C and a functional form extracted from the data f' (an approximation of the function f that we want to learn), the correct logic formulation for this check is $f' \rightarrow C$. This means checking

if a property C is satisfied by f' . You can see this formulation at page 1 of LGML paper (using the entailment symbol). In LGGA this consistency check ($f' \rightarrow C$) is done in two points of their pipeline: 1) In the loss when computing the truth error (line 3 of Algorithm 1, page 4): they check if the constraints are satisfied by the function f' ($f' \rightarrow C$) and if not, they compute the magnitude of the truth error and they incorporate it in the loss. 2) When they compute the counterexample points (line 6 of algorithm 1, page 4). Here they use the AT to generate new data points. This is possible only when f' does not satisfy the constraints ($f' \rightarrow C$ fails). The point 2) is the same as in LGML.

In our method not only are the constraints fundamentally different (as explained above), the logic statement we want to prove is radically different: We want to prove that $B \rightarrow f'$, where B is a collection of axioms. That means that we can deduce the formula f' from a set of physics laws. This is very different from LGGA and LGML where the objective is to check if a single property is satisfied by a given formula f' . In formal logic, these are two fundamentally different tasks: The first one (LGGA & LGML) is *consistency checking*, the second one (AI-Descartes) is *logical deduction*. Thus our work is not very closely related to LGGA/LGML, as these approaches (LGGA/LGML and ours) solve different logical problems.

Does the work support the conclusions and claims, or is additional evidence needed?

All claims seem correct. However, it is not clear if its a state-of-the-art tool for symbolic discovery. Based on the empirical evaluations of tools like AI-Feynman and PySR, it seems these tools would also be able to solve the underlying symbolic regression problems considered in the paper. Is this not the case? What is the value-addition of AI-Decartes here?

>>> **Our pipeline is focused on real datasets (few datapoints and noisy). The 3 examples we show in the paper (6 datasets) are showcases of such datasets. We can see that (in the section "Comparison with other systems" in the supplementary material, Tables 7-12 and Figures 5-8) when running AI-Feynman and PySR on our 6 real-measurements dataset, their performance is quite poor compared to synthetic data: the systems very rarely produce the correct formula as the best candidate. Moreover, when considering the AI-Feynman dataset (Tables 13-14) we obtain similar results, outperforming all the systems with an improvement of at least 10%.**

Please see also our general response Q1 and Q2 and Q3 and the new set of SR experiments in Tables 13-14 in the supplementary material.

The authors go into great depth on three examples and have made a case that the idea of the paper has some merit on these examples. However, just three benchmarks remain unjustified. The authors make a note of other tools such as AI-Feynman PySR but do not baseline against it. AI-Feynman provided a significant dataset of several physics problems. There is a benchmark suite by William La Cava's group (<https://github.com/cavalab/srbench>). Furthermore, there is now a competition in this space at GECCO 2022.

>>> **We now added a comprehensive comparison with the state-of-the-art tools (AI-Descartes, AI-Feynman, PySR, Bayesian Machine Scientist and TuringBot) on the**

AI-Feynman dataset, showing that we outperform them with an improvement of at least 10%.

Please see our previous response for more details.

Are there any flaws in the data analysis, interpretation and conclusions? Do these prohibit publication or require revision?

There are several modules in the figure architecture diagram that remain either unexplained or have an unclear impact.

What is the experimental design module? There is almost no commentary on this.

However, it is an integral part of the tool. It seems similar to the Oracle Labeler in Scott et al. ?

The oracle labeler is a major assumption made by that tool. This needs to be expanded on.

>>> In this study we elected to showcase the capability of the two main components of the pipeline (i.e. deductive reasoning module and symbolic regression) and to exclude details regarding the dotted parts in Figure 3 of the main paper.

It is important to note that all the experiments shown in the paper are conducted exclusively by using the two modules of deductive reasoning and symbolic regression. We did not employ any tool indicated by dotted boundaries in the system figure (Fig 3 in the main part of the paper) such as abductive reasoning, model refinement, experimental design, etc.

We present the envisioned big picture primarily to provide broader context with regard to the holistic view of the discovery cycle, as well as to encourage more researchers to further contribute. However, we have developed an experimental design module, but this goes beyond the scope of this work (see references below).

Although optimal experimental design (OED) has similarities with the Oracle Labeler in Scott et al. in that both choose data points, the two are quite different in purpose and implementation; as discussed in detail below, OED is concerned with finding the data point that is most informative in distinguishing among different models or their parameterization, where in the typical Bayesian setting, the state of belief about the models is governed by a probability distribution over the models, including both their functional form and the parameters associated with each functional form. Based on this distribution, and a distribution over measurement noise, a data point is found such that the response data measurement, once taken, will sharpen the model distribution. The Oracle Labeler, on the other hand, is invoked when a known constraint on a model (such as symmetry in its input arguments) may be violated; the Labeler returns a label for the identified counterexample datapoint (which corresponds to the value of the function evaluated on the counterexample). The current candidate models will then perform poorly on this new datapoint and will thereby guide the next iteration of symbolic regression to avoid such models. Since the function is not known, we cannot be sure that the oracle will produce a sound data point.

While a Bayesian prior distribution could in principle capture symmetry and other simple constraints (by giving them a prior probability of zero), and thereby subsume the role of the Oracle Labeler, this is not the primary goal of OED.

The experimental design module addresses the fundamental question of devising what data should be collected, so as to best inform the discovery process. This question is particularly important, in situations where data are scarce, expensive to acquire, or restricted. The field of experimental design (and its more recent variant, active learning) has been advancing steadily over the years drawing insights from both probability theory as well as information theory. Most literature divides into two variants of the problem. The first variant is model selection, that is, given a plurality of models of different functional forms, devise an experiment whose outcome will help probabilistically, to substantiate the choice of either of the models as more likely (e.g. imagine we have two simple models $y_1=x^2$ and $y_2=0.5*\exp(x)-1$, the two may fit noisy data at the domain of $[0, 2]$ comparatively, yet, if we can determine for what value of x we should consider for an experiment that will tell whether one model is more likely than another, probably a choice of the largest x within the domain, would maximize the divergence between the predictions of the two models). The other variant, often implicitly presumes a given functional form for a model $y=(x;z)$ that can be tuned by some parameters z , where the objective is to propose experiments (here x values) whose outcomes (y 's) best inform the distribution of the underlying model's parameters (e.g. determine sensor placements, such that the observations help to substantiate some underlying model parameters z). In the discovery context, we are provided with the choice to determine both the functional form, and the underlying parameters, therefore, an experimental design framework that can attend to both considerations is essential. On a high level the OED problem can be articulated as a meta-level optimization problem, where the objective consists of some utility function that attends to an information theoretic measure of the reduction in uncertainty that the experiment is expected to yield. Given the non-trivial scope of the OED module, we provide reference to our patent, specifically addressing OED in the context of our discovery framework, "*Experimental Design for Symbolic Model Discovery*". L. Horesh, K. Clarkson, C. Cornelio, S. Magliacane, filed on: April 2020 (<https://patents.google.com/patent/US20210334432A1>) as well as a reference to our preprint description ("*Bayesian Experimental Design for Symbolic Discovery*", Clarkson et al. <https://arxiv.org/abs/2211.15860>), yet we elected to keep the narrative focused on the integration of reasoning and symbolic regression, rather than potentially overloading the readers with several narratives.

The abductive reasoning module sounds quite interesting, but it is unclear how it works from the paper. What "new axioms" did it propose in the three experiments? Did it find anything useful?
>>> We did not integrate the abductive reasoning module in the current work. This work focuses on the interplay of the symbolic regression module and the deductive part of the reasoning module.

All the experiments and the definitions in this paper do not involve the abductive reasoning module.

However, we show the envisioned big picture and full capabilities of our pipeline to inspire more researchers to contribute.

Abductive reasoning is a well-studied research topic in the logic domain. Many abductive reasoning tools are freely available and are suitable to be integrated in our pipeline. For simplicity, in this work we make the assumption of complete and consistent background theory. In this scenario there is no need for logic abduction because, given the completeness assumption, we already have all the axioms needed to prove the formulas. We described what abductive reasoning is in the section “System limitations and future improvements” in the Method part of the paper.

For those unfamiliar with the topic, we added a pointer to an overview page, the well curated entry of the Stanford Encyclopedia of Philosophy on abductive reasoning <https://plato.stanford.edu/entries/abduction/>.

Is the methodology sound? Does the work meet the expected standards in your field?

The authors go into great depth on three examples. While I appreciate the author's argument in the rebuttal for only considering 3 benchmarks, it is still unclear to me if there is publication bias. Even just a few additional experiments (even if not in as such detail as before) would help. I am unable to draw any conclusions on how AI-Decartes compares to state-of-the-art tools in this space. There is also a benchmark suite provided by ca Furthermore, there have been recent competitions in this space,

>>> **Other than the 6 original datasets we considered (for the 3 examples), we added more experiments to compare with the state-of-the-art. See answer to Q2 of our general response for more details.**

However, the prerequisites for our model are the availability of a formalized background theory and a corresponding dataset with data collected by real-world measurements. Unfortunately, this type of data currently is still not easy to find. See answer Q4 for more details.

Is there enough detail provided in the methods for the work to be reproduced?

A link to a github source code is provided in the rebuttal but not in the paper.

>>> **Thanks for noticing. We added the github link to the paper (see section “Data and Code availability” in the Methods Part of the paper, before the references).**

Why is the number of const part of the input? This is atypical for symbolic regression problems.

>>> **We brought this in to reflect a heuristic in fitting equations to data in engineering practice. Expressions with more fitted parameters generally require more data to fit than expressions with fewer. Moreover, accounting for complexity through expression length alone doesn't capture the apparent increase in complexity from an expression with a constant c_1 in multiple positions (e.g. $c_1*p + c_1*p^2$) and an expression with equivalent structure but with two unique constants ($c_1*p + c_2*p^2$). We considered the distinction between these to be useful in our investigation, as many theory-derived expressions reuse meaningful constants.**

What is the experimental design module? There is almost no commentary on this. However, it is an integral part of the tool. It seems similar to the Oracle Labeler in Scott et al. ? The oracle labeler is a major assumption made by that tool. This needs to be expanded on.

>>> See the detailed answer on experimental design above.

Summary

Overall, I feel like the system behind AI-Descartes shows great promise. However, I agree with reviewers #2 and #3 that the empirical evaluation is unconvincing. It is not clear to me where the tool falls relative to state-of-the-art, and several components of the tool are not discussed, and the value-addition is unclear. The idea of the paper is quite similar to Scott et al. and Ashok et al., while there are some differences, the authors' explanation of the difference is not clear and believe to be incorrect.

REVIEWERS' COMMENTS

Reviewer #4 (Remarks to the Author):

The authors have responded and addressed most of my concerns.

The new results on the AI-Feynman dataset is extremely convincing. I hope to see it competing and succeeding against other state-of-the-art tools in competitions!

My biggest pushback with the paper in its current form is the presentation of the figure architecture diagram. While it is now clear that the dotted lines are hypothetical, I would perhaps find a way to put more emphasis on it.

TO THE REVIEWERS:

We sincerely thank the referees for carefully reviewing the manuscript and helping us improve it further.

As suggested by Reviewer 4 we have modified the architecture diagram by adding a more obvious way to distinguish between the components described in detail in the paper and the ones that are not. We changed the color to all the additional components in gray and we added a corresponding description to the caption of the figure.

We agree that the new architecture figure is now easier to interpret.

Reviewer #4 (Remarks to the Author):

The authors have responded and addressed most of my concerns.

The new results on the AI-Feynman dataset is extremely convincing. I hope to see it competing and succeeding against other state-of-the-art tools in competitions!

My biggest pushback with the paper in its current form is the presentation of the figure architecture diagram. While it is now clear that the dotted lines are hypothetical, I would perhaps find a way to put more emphasis on it.

>> Done (see above).